# GAUSSIANTRIM3R: CONTROLLABLE 3D GAUSSIANS PRUNING FOR FEEDFORWARD MODELS

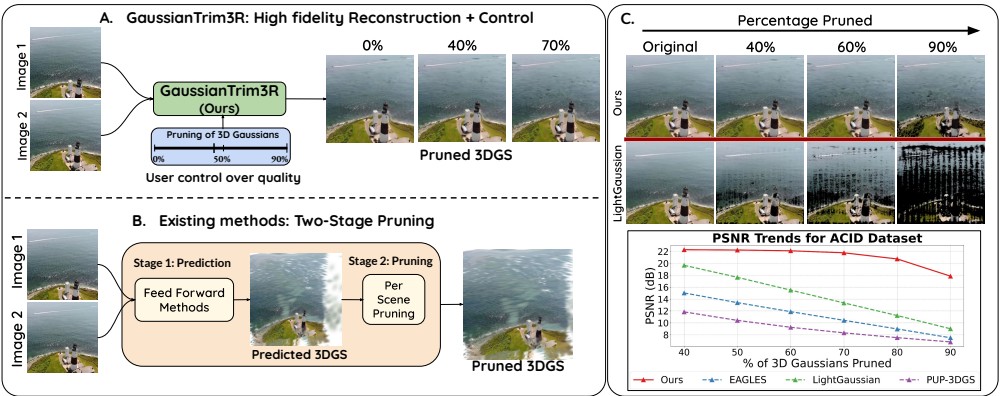

Figure 1: **GaussianTrim3R:** We highlight **GaussianTrim3R**'s ability to *maintain scene quality at high pruning rates*. **(Left)** Schematic comparison of our approach with conventional methods, emphasizing how our texture-aware pruning preserves scene fidelity at high sparsity. **(Right)** Qualitative results and PSNR trends of our approach demonstrating superior novel view synthesis quality even when pruning up to 90% of Gaussians.

## ABSTRACT

Feed-forward methods offer a promising paradigm for novel-view synthesis, eliminating computationally expensive per-scene optimization. However, current feed-forward approaches typically predict a fixed number of pixel-aligned Gaussian primitives, leading to significant redundancy. Naively pruning these Gaussians creates severe visual artifacts, necessitating fine-tuning that compromises the feed-forward nature. We introduce **GaussianTrim3R**, a novel framework for controllable and feed-forward 3D Gaussian representation method which gradually prunes 3D Gaussians and simultaneously adjusts the attributes of remaining Gaussians maintaining rendering quality, thus eliminating the need for finetuning 3D Gaussians post pruning. To achieve this, we construct SuperClusters by partitioning the 3D scene based on spatial and color attributes. By leveraging Discrete Wavelet Transform, we assign and rank texture complexity to these SuperClusters, enabling selective, texture-aware pruning. Doing so enables our method to directly predict attribute-adjusted Gaussians, thereby preserving scene integrity. Unlike existing methods, GaussianTrim3R offers an efficient, real-time solution with extensive experiments demonstrating superior trade-offs between quality and efficiency across diverse real world RealEstate10K, ACID and DTU datasets.

## 1 INTRODUCTION

Sparse-view 3D reconstruction, a cornerstone challenge in computer vision, underpins critical applications in robotics, AR/VR, and digital twin creation. Inferring dense 3D structure from limited observations is inherently ill-posed. While recent advancements like NeRF Mildenhall et al. (2020) and 3D Gaussian Splatting (3DGS) Kerbl et al. (2023a) achieve remarkable photorealistic synthesis, their reliance on dense views and per-scene optimization limits adaptability and efficiency in dynamic, real-time environments. Feed-forward methods Wang et al. (2024); Leroy et al. (2024) offer a

**Pruning on low texture regions**        **Optimizing remaining Gaussians**

Figure 2: Here we show a toy experiment demonstrating our core insight for GaussianTrim3R. We render an image of a scene after selectively pruning 3D Gaussians. (**Left**) A rendered view shows visible artifacts, such as "black patches," that are indicative of naive pruning in low-texture regions. (**Right**) The same scene is rendered after fine-tuning the remaining Gaussians on the ground-truth image. The rendering quality is significantly restored because the surviving Gaussians in the low-texture areas have adaptively expanded to fill the gaps, thereby eliminating the observed artifacts. This experiment underscores that texture-aware pruning, combined with adaptive reallocation, is crucial for maintaining scene fidelity under aggressive pruning. Our work, **GaussianTrim3R**, exploits this very insight in feed-forward manner, eliminating the need for computationally expensive per-scene optimization.

compelling alternative, directly inferring 3D representations with superior efficiency and generalization across diverse scenes, bypassing the computationally expensive steps of Structure-from-Motion (SfM) and radiance field optimization. Despite their efficiency, existing feed-forward approaches typically infer a fixed quantity of pixel-aligned 3D Gaussians regardless of scene complexity. This architectural rigidity often leads to redundant representations and a fundamental inability to control quality. Some methods even impose stricter constraints Smart et al. (2024); Ye et al. (2024), like one Gaussian per pixel, further restricting adaptive scene representation. This lack of dynamic Gaussian allocation makes existing feed-forward unsuitable in resource constrained real-time applications where optimially allocating 3D Gaussians are essential.

A naive random pruning attempt to address these inefficiencies inevitably leads to severe artifacts in novel views. Such aggressive pruning creates undesirable empty regions within the 3D scene, disrupting continuity and severely degrading rendering quality. While compression techniques exist Girish et al. (2024); Morgenstern et al. (2025); Fang & Wang (2024); Ali et al. (2024); Navaneet et al. (2023); Niedermayr et al. (2024), they primarily focus on post-hoc quantization or encoding, typically requiring dense multi-view inputs and scene-specific optimization. *This dependence inherently limits their adaptability and applicability to the challenging sparse-view feed-forward paradigm.* Alternatively works like Tang et al. (2024a) hierarchically densifies 3D Gaussians across the scene by promoting inter-scale interaction but doesn't provide pruning technique with fine grain control, similarly Hong et al. (2024) directly estimates pixel aligned 3D Gaussian with no flexibility on representation. These critical limitations underscore an urgent need for a more adaptive, feed-forward, and user-controllable approach that not only enhances rendering efficiency but also provides finegrained control over 3D Gaussian pruning based on scene complexity.

In this work, we introduce **GaussianTrim3R**, a novel, feed-forward controllable 3D Gaussian allocation method designed to provide real-time control over the number of 3D Gaussians while maintaining rendering quality and computational efficiency. Our core insight is rooted in the predictable statistical properties of natural scenes: their Power Spectral Density (PSD) follows a power-law decay ($PSD \propto \frac{1}{f^\alpha}$ where $f$ is frequency and $\alpha \approx 2$) with spatial frequency Torralba & Oliva (2007). Since sparse-view settings inherently lack the dense multi-view signals typically used for pruning, we leverage this natural scene statistic to guide our approach. The Discrete Wavelet Transform (DWT) is an effective tool for analyzing local frequency content and is known to capture this power-law behavior Pando & Fang (1998) in natural images. Specifically, we employ the DWT to identify smooth regions which are candidates for 3D Gaussians reductions. Our strategy involves omitting 3D Gaussians in these low-texture (smooth) areas while simultaneously replacing them with fewer but larger 3D Gaussians to maintain scene quality. As shown in Figure 2, a naive pruning approach introduces visual artifacts, but subsequent Gaussian adjustments quickly recover high-quality rendering. The remaining Gaussians in these regions adaptively grow to fill the vacated space, demonstrating that a well-designed allocation strategy is key to preserving fidelity.

We extend this principle to sparse-view feed-forward methods. **GaussianTrim3R** performs gradual, texture-aware pruning by segregating 3D regions based on their textural complexity, targeting low-texture areas for simplification. By eliminating the need for per-scene optimization, our method is the first to perform 3D Gaussian pruning in feed-forward manner, a significant step toward real-time, efficient rendering.

To ensure scene continuity and mitigate potential artifacts like holes or gaps (especially during aggressive pruning), we introduce an adaptive Gaussian expansion mechanism. This mechanism dynamically adjusts the remaining Gaussians to provide comprehensive coverage of the simplified regions, while maintaining its feed-forward nature. Our method consistently maintains high novel view quality, even under high pruning rates, outperforming existing approaches which suffer invariably from quality degradation. Furthermore, upon rendering, this results in preserving crucial scene structure, thereby supporting improved scene understanding.

Our approach is summarized in Figure 1. It offers both qualitative and quantitative improvements over existing baselines, especially in high pruning scenarios. As can been seen in the figure, even at a high pruning rate of $90\%$, GaussianTrim3R retains remarkable scene detail, making it invaluable for various applications. Our contributions can be summarized as follows:

- We propose **GaussianTrim3R**, a novel, feed-forward method for dynamic, global control over the number of 3D Gaussians at inference time, enabling real-time trade-offs between rendering quality and computational efficiency.

- We introduce a Texture-aware selective pruning strategy that prioritizes Gaussian reduction in less critical, low-detail areas, thereby preserving high-detail regions and significantly boosting efficiency.

- We also propose an Adaptive Gaussian expansion mechanism that allows remaining Gaussians to dynamically grow and cover pruned areas, effectively preventing holes and discontinuities in the reconstructed scene.

- We demonstrate state-of-the-art Sparse-View 3D reconstruction results on datasets like RE10K, ACID and DTU as measured by metrics like PSNR, LPIPS, and SSIM. Further, we show that even at extremely high pruning rates of up to $90\%$, we main high-fidelity of the scene, achieving improvement of $8dB$ in PSNR in such conditions compared to baselines.

## 2 Related Work

### 2.1 Radiance Fields and Sparse View Reconstruction

Reconstructing 3D scenes from multi-view images is a core computer vision problem. Modern radiance field methods like NeRF Mildenhall et al. (2020) and 3D Gaussian Splatting (3DGS) Kerbl et al. (2023a) achieve high-quality novel view synthesis, with 3DGS offering notable advantages in rendering speed through rasterization. These approaches have found success in applications such as 3D asset generation Liu et al. (2023); Lin et al. (2025); Meng et al. (2024); Gao* et al. (2024); Zhou et al. (2024a;c); Kim et al. (2024); Yang et al. (2024), scene editing Haque et al. (2023); Wu et al. (2024a); Fang et al. (2024); Srinivasan et al. (2021); Jiakai et al. (2021); Hyung et al. (2023); Dong & Wang (2025); Chen et al. (2024a); Yao et al. (2024), and dynamic scene reconstruction Kerbl et al. (2023b); Luiten et al. (2024); Wu et al.; LIU et al. (2025); Pumarola et al. (2020).

However, they often require dense multi-view images. To address this, sparse-view reconstruction methods Yu et al. (2021); Charatan et al. (2024); Wewer et al. (2024); Liu et al. (2023); Wu et al. (2024b); Poole et al. (2022); Koo et al. (2024) leverage learned priors or foundation models Rombach et al. (2022). More recently, feed-forward approaches like DUSt3R Wang et al. (2024), MASt3R Leroy et al. (2024) and VGGT Wang et al. (2025a), built on ViT backbones, directly predict dense 3D scene representations from limited views Ye et al. (2024); Hong et al. (2024); Xu et al. (2024); Wang et al. (2025b); Nam et al. (2024); Keetha et al. (2024). These scalable methods have been extended to dynamic scenes Zhang et al. (2024), multi-view consistency Asim et al. (2024), and estimating 3DGS representations Zhou et al. (2024b); Yu et al. (2024); Ye et al. (2024); Smart et al. (2024); Tang et al. (2024b), even from just a few hundred views Wang et al. (2025a).

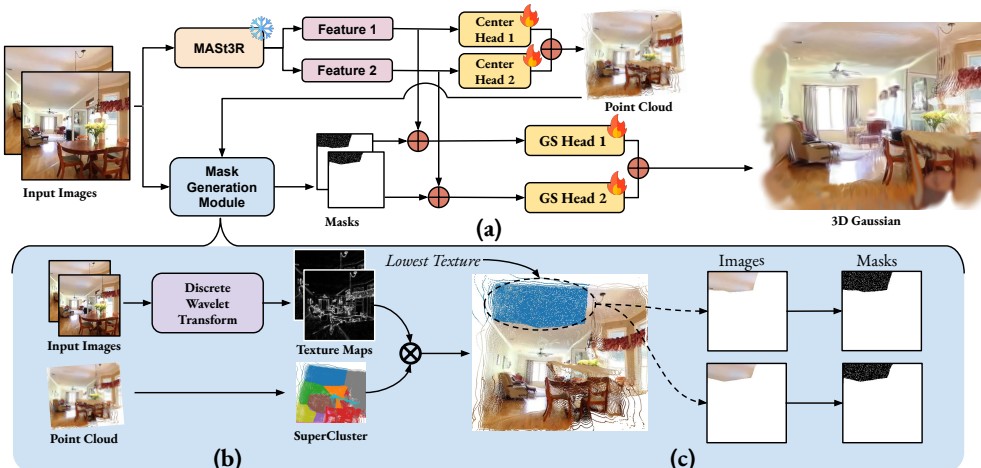

Figure 3: Our pipeline dynamically allocates a budget of 3D Gaussians for real-time, novel-view synthesis. (a) **Initial Representation:** We infer a dense 3D point cloud from context images using the MASt3R backbone. (b) **SuperCluster Formation and Texture Analysis:** The point cloud is partitioned into SuperClusters, and texture information is simultaneously analyzed using DWT to identify low-texture regions. (c) **Texture-Aware Pruning and Adaptive Allocation:** Based on the analysis, our system generates masks that guide a Gaussian Head to predict optimized parameters for the remaining, larger Gaussians, yielding a high-quality, sparse 3D representation.

## 2.2 COMPRESSION IN 3D GAUSSIAN SPLATTING

Due to its explicit nature, 3DGS often requires a large number of Gaussians to represent a scene, leading to higher storage, processing time, and slow rendering. To address these challenges, several works focus on compressing 3DGS representations. EAGLES Girish et al. (2024) and Light-Gaussian Fan et al. (2023) primarily use quantization of attributes and pruning based on Gaussian contribution, PUP-3DGS Hanson et al. (2025) uses uncertainty estimation to estimate Gaussians importance, and SOG-3DGS Morgenstern et al. (2025) applies image compression techniques. Other methods like Ali et al. (2024) trim Gaussians with low gradients followed by fine-tuning, while Fang & Wang (2024) proposes pruning and densification based on alpha blending importance. However, a common limitation across these compression techniques is their reliance on quantization or their inability to dynamically modify the attributes of remaining Gaussians post-pruning, which can lead to visible artifacts. Moreover all existing method require dense multi-view images of the scene to compress which fails in sparse view feed forward setting. Interestingly previous work like Xu et al. (2023) have explored to use wavelet based frequency decomposition to achieve high quality synthesis for multiview images but no such method exists for 3D Gaussian Splatting representation

## 3 PROPOSED METHOD

We now present **GaussianTrim3R**, a novel controllable sparse view feed-forward 3D Gaussian pruning pipeline, illustrated in Fig. 3.

**Objective.** Our primary objective is to prune 3D Gaussians to given a budget of $Z$ 3D Gaussians. We achieve this by strategically identifying and processing regions suitable for efficient 3D Gaussian allocation while pruning rest, using sparse-view images $I_1, I_2 \in R^{H \times W \times 3}$, where $H$ and $W$ are the height and width of the images. To do this, we generate masks $M_1, M_2 \in R^{H \times W}$, that indicates which pixel-aligned 3D Gaussians should be used for scene representation. These masks are concatenated with decoder features of MASt3R and fed into a Gaussian Head $f_\theta$, which predicts the required number of 3D Gaussians. Our core insight is to target low-texture regions, where dense pixel-aligned 3D Gaussian primitives are replaced by a much smaller set of larger, size-adapted 3D Gaussians. This approach is effective because natural images Power Spectral Density (PSD) follows power law decay with spatial frequency.

To generate mask we first detail how we partition the regions within a 3D scene to form SuperClusters in Sec. 3.1.1 and allocation of 3D Gaussian within these SuperClusters based on the proposed measure of "textureness" and we elaborate on usage of mask to predict adapted 3D Gaussians in Sec. 3.1.2. We follow that with description of our overall training pipeline in sec 3.2.

## 3.1 MASK GENERATION

### 3.1.1 SCENE PARTITIONING: SUPERCLUSTERS FORMATION

We apply $K$-means clustering to a pointmap $P \in R^{2HW \times 6}$ obtained from MASt3R. This process aggregates points with similar spatial and color attributes into distinct clusters: *SuperClusters*. Each point $p \in R^6$ is represented by its position $(x, y, z)$ and its assigned RGB color values. A SuperCluster, $SC_i$, is a subset of the pointmap where every point in $SC_i$ is closer to the cluster's centroid $c_i$ than to the centroid of any other SuperCluster. Formally,

$$SC_i = \{p \in P; ||p - c_i||^2 \leq ||p - c_j||^2, \forall j \in \{1, 2, ..., K\}, j \neq i\} \tag{1}$$

where, centroid of a SuperCluster is calculated as the mean of its points; $c_i = \frac{1}{|SC_i|} \sum_{p \in SC_i} p$

### 3.1.2 TEXTURE-AWARE PRUNING WITHIN SUPERCLUSTERS

Once the SuperClusters are obtained, the pivotal step is to imbue each with a robust measure of its texture complexity.

**Quantifying Textureness of SuperClusters.** We quantify the texture of each SuperCluster by first calculating the local texture energy for each 3D point's corresponding pixel in the context views, $I_1$ and $I_2$. We apply the Discrete Wavelet Transform (DWT) Mallat (1989) to context images to capture localized frequency variations. We use Daubechies wavelet coefficients, which provide a robust directional frequency decomposition across multiple scales. To measure texture energy, we compute the absolute sum of coefficients instead of the conventional squared sum, which is more robust and less sensitive to outliers. The local texture energy, $E_{i,j}$, for a given pixel $(i, j)$ for each context views is defined as the sum of the absolute values of its wavelet coefficients across all decomposition directions: $E_{i,j} = \sum_{k \in \{h,v,d\}} |W_{ij}^k|$, where $W_{ij}^k$ denotes the wavelet coefficient and $\{h, v, d\}$ represents horizontal $(h)$, vertical $(v)$ and diagonal $(d)$ directions, respectively, at pixel $(i, j)$ and $D$ is the total number of decomposition directions ($D = 3$, horizontal, vertical and diagonal directions)

Finally, we quantify the texture of each SuperCluster by averaging the $E_{i,j}$ values of all 3D points within it using pointmap representation. This allows us to quantitatively rank SuperClusters based on their collective texture complexity, which forms the basis for 3D Gaussian allocation and pruning.

$$\text{Texture}(SC_i) = \frac{1}{|SC_i|} \sum_{p \in SC_i} E_{\pi(p)}; \pi : R^3 \to \mathbb{Z}^2, \tag{2}$$

where $\pi$ maps a 3D point to a pixel coordinate in one of the context views.

**Ranking SuperClusters and Gaussians allocation** In this stage, we create a mask for each context view's decoder feature, that specifies which 3D Gaussians will be used to represent the scene and which will be pruned. To do this, we first rank all SuperClusters, $SC_i \forall i \in \{1, 2, .., K\}$, based on their texture score, $\text{Texture}(SC_i)$, from lowest to highest. We then process these clusters iteratively, starting from the lowest-textured one, until our predefined global 3D Gaussian budget, $Z$, is met. For each $SC_i$ selected, we retain a percentage ($N\%$) of its points, which are uniformly spread throughout the cluster and prune the rest. This ensures we can effectively cover the entire low-texture region with 3D Gaussians with only $N\%$ and pruning rest. We empirically choose $N = 10\%$ for our method (see sec A.2 for additional experiments with different values of $N$). We predict 3D Gaussians only for these retained points. We create binary masks, $M_1$ and $M_2$, for each context view to denote which points from the point map, $P$, are retained. These masks are then used as input to the Gaussian Head, along with decoder features, to predict the 3D Gaussians. Let $N_{SC_i}$ be the number of points retained from $SC_i$. The total number of retained points must equal the budget $Z$ i.e. $\sum_{i=1}^{K} N_{SC_i} = Z$. We want the 3D Gaussians predicted for the retained points in low-texture SuperClusters to be larger. This is because these regions are smooth and can be effectively covered by a smaller number of larger Gaussians.

| Pruning Method | 78k Gaussians | | | 52k Gaussians | | | 13k Gaussians | | |
|---|---|---|---|---|---|---|---|---|---|
| | PSNR↑ | LPIPS↓ | SSIM↑ | PSNR↑ | LPIPS↓ | SSIM↑ | PSNR↑ | LPIPS↓ | SSIM↑ |
| EAGLES | 13.769 | 0.483 | 0.461 | 10.908 | 0.588 | 0.289 | 6.591 | 0.701 | 0.076 |
| LightGaussian | 18.218 | 0.357 | 0.621 | 13.642 | 0.486 | 0.444 | 7.513 | 0.684 | 0.123 |
| PUP-3DGS | 11.649 | 0.396 | 0.510 | 8.901 | 0.490 | 0.352 | 5.823 | 0.647 | 0.088 |
| EfficientGS | 13.755 | 0.407 | 0.520 | 10.686 | 0.513 | 0.384 | 6.329 | 0.682 | 0.116 |
| Ours | **22.294** | **0.235** | **0.735** | **22.110** | **0.243** | **0.726** | **17.848** | **0.314** | **0.645** |

Table 1: **Results on RE10K dataset** with existing pruning methods **without finetuning the remaining Gaussians.** We see baselines performance degrades with increase in pruning strength. We observe absence of 3D Gaussians in regions of scene. In extreme pruning scenraio, our method outperforms the baselines by a large margin.

| Pruning Method | 78k Gaussians | | | 52k Gaussians | | | 13k Gaussians | | |
|---|---|---|---|---|---|---|---|---|---|
| | PSNR↑ | LPIPS↓ | SSIM↑ | PSNR↑ | LPIPS↓ | SSIM↑ | PSNR↑ | LPIPS↓ | SSIM↑ |
| EAGLES | 11.047 | 0.600 | 0.343 | 10.786 | 0.623 | 0.315 | 9.626 | 0.688 | 0.217 |
| LightGaussian | 10.992 | 0.609 | 0.326 | 10.881 | 0.616 | 0.312 | 10.077 | 0.661 | 0.241 |
| PUP-3DGS | 9.672 | 0.640 | 0.266 | 8.616 | 0.673 | 0.204 | 7.261 | 0.722 | 0.101 |
| EfficientGS | 9.171 | 0.666 | 0.193 | 9.290 | 0.661 | 0.196 | 9.038 | 0.676 | 0.182 |
| Ours | **22.549** | **0.299** | **0.640** | **22.310** | **0.310** | **0.627** | **18.476** | **0.379** | **0.553** |

Table 2: **Results on ACID dataset** with existing pruning methods on feed forward backbone, where **we finetune the remaining Gaussians** to maintain the scene continuity.

## 3.2 Overall Training pipeline

To ensure that the model's performance is stable across diverse budget $Z$ rates at test time, it is important that it generalizes well to a broad spectrum of pruning percentages during training. To achieve this, we train the Gaussian Head with various budget during training and execute our texture-aware allocation strategy on the SuperClusters with that budget. This strategy improves robustness of the method to different budget $Z$, enabling adaptation of the retained 3D Gaussians at test time.

The network is trained by comparing the rendered 2D image generated from the retained 3D Gaussians at a specific target camera pose, with the corresponding ground truth image using photometric loss. Optimization is performed using a combination of Mean Squared Error (MSE) and LPIPS Zhang et al. (2018) loss to ensure high visual fidelity. The overall loss formulation $\mathcal{L}$ is as follows:

$$\mathcal{L} = \text{MSE}(\hat{I}_v, I_v) + \lambda\text{LPIPS}(\hat{I}_v, I_v) \tag{3}$$

Here $v$ represents the target camera view direction, $I_v$ denotes the ground truth target image and $\hat{I}_v$ denotes the rendered image from the target view $v$ of 3D Gaussians obtained from Gaussian Head $f_\theta$ using context view decoder features along with its corresponding masks $M_1, M_2$. $\lambda$ controls the regularization strength, which is set to $0.001$ in all our experiments.

## 4 Experiments

**Implementation Details.** Our approach builds upon the architecture of DUSt3R Wang et al. (2024) with a Gaussian Head to predict 3D Gaussian primitives similar to Ye et al. (2024), which integrates a Gaussian prediction head into the MASt3R Leroy et al. (2024) pipeline. This foundational design allows for the direct extraction of 3D Gaussian attributes for each point in the point cloud obtained from MASt3R, providing a robust initial 3D representation. We choose $K = 300$ and $N = 10\%$ for our pipeline, we provide ablation on these hyperparameters in sec 4.2.

**Datasets.** We train and evaluate our method on diverse real-world datasets following the experimental setup of pixelSplat Charatan et al. (2024), MVSplat Chen et al. (2024b), and NoPoSplat. Our primary training and evaluation data is derived from RealEstate10K Zhou et al. (2018) (indoor, multi-view videos) and ACID Xiao & Kang (2021); Xiao et al. (2022) (large-scale, dynamic outdoor drone scenes) datasets. Both these datasets provide multi-view images with COLMAP-computed Schönberger et al. (2016) camera poses, and we adhere to their official train-test splits. To

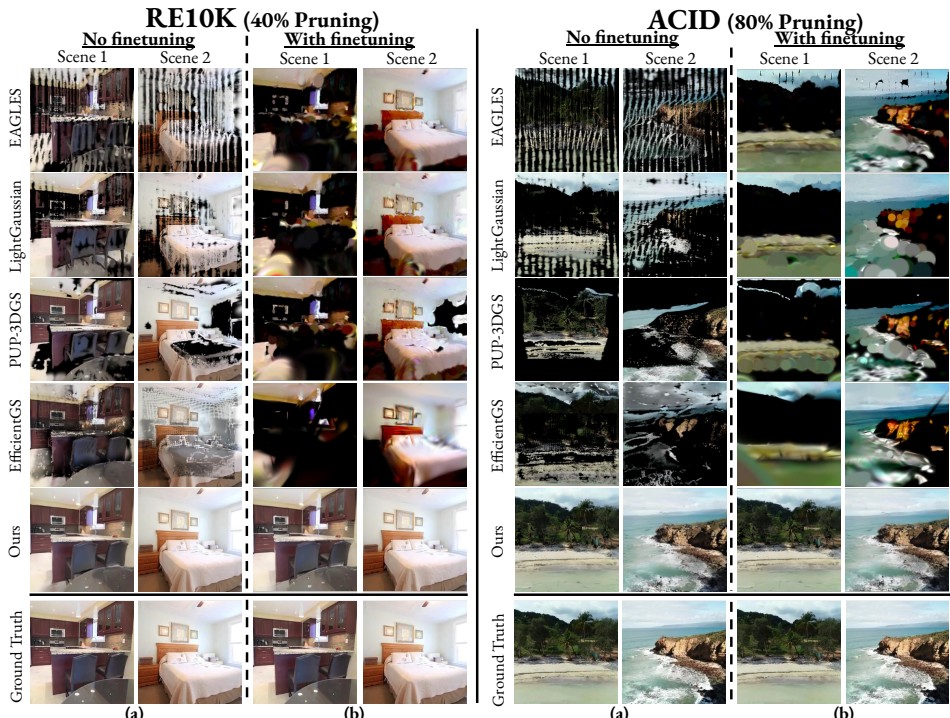

Figure 4: Comparison of baseline results on RE10K and ACID scenes using $40\%$ and $80\%$ of 3D Gaussians respectively. We allocate fewer but larger 3D Gaussians to low texture SuperClusters by the feedforward backbone.

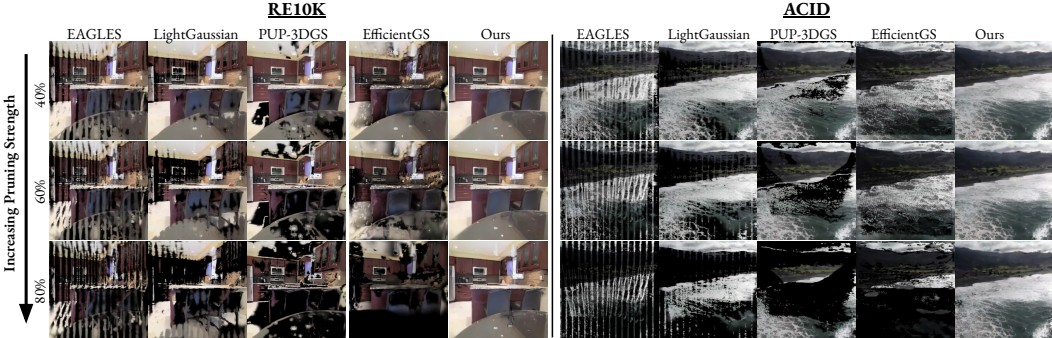

Figure 5: **Qualitative comparison** of our method with baseline methods (no finetuning) on a single scene from RE10K and ACID dataset. Our method faithfully represents a scene even with a tight budget ($80\%$ pruning).

assess zero-shot generalization, we also evaluate on the DTU dataset Jensen et al. (2014), comprising high-quality, object-centric scans.

**Evaluation Criteria.** We benchmark our method on novel view synthesis using widely adopted image quality metrics: Peak Signal-to-Noise Ratio (PSNR), Structural Similarity Index Measure (SSIM) Wang et al. (2004), and Learned Perceptual Image Patch Similarity (LPIPS) Zhang et al. (2018). Beyond these measures, we explicitly compare against the no. of Gaussians used for 3D scene representation. This metric directly demonstrates our approach's efficiency in reducing redundancy while maintaining superior visual fidelity.

**Baselines.** Direct comparison with existing 3D Gaussian pruning methods is challenging, as they typically operate per-scene and are optimized for dense multi-view inputs, unlike our generalized,

feed-forward approach. To enable meaningful comparison, we construct baselines by applying established pruning strategies on top of state-of-the-art feed-forward 3D Gaussian generation backbones like pixelSplat, MVSplat, NoPoSplat. Specifically, from the Gaussians generated by these backbones, we apply four distinct pruning strategies: **(1) Random Pruning:** Gaussians are randomly removed to meet a target count or percentage; **(2) EAGLES:** Prunes based on scene-wide influence, targeting low-contribution Gaussians (e.g., small scale, low opacity, occlusion); **(3) Light-Gaussian:** Prunes based on estimated importance (ray hits, volume); **(4) PUP-3DGS:** It assigns a pruning score by computing second order approximation of reconstruction error on input views with parameters of Gaussians and **(5) EfficientGS Liu et al. (2025):** It estimates importance of Gaussians per pixel by selecting Top-K Gaussians to be retained. For each baseline, we present results under two scenarios: **(1) Direct pruning and rendering**, maintaining feed-forward nature, and (2) **Fine-tuning the remaining Gaussians** for a few iterations per scene. Note that finetuning breaks the inherent feed-forward nature by necessitating additional per-scene optimization, underscoring a key advantage of our proposed GaussianTrim3R, which operates entirely feed-forward.

## 4.1 RESULTS

**Quantitative Evaluation.** We provide a comprehensive evaluation of our method's performance on the novel view synthesis task employing standard metrics: PSNR, LPIPS, and SSIM. Table 1 presents a detailed comparison of GaussianTrim3R against established baselines on the RE10K dataset without any finetuning. and Table 2 presents results on ACID dataset with finetuning. Performance is reported across three different pruning levels, corresponding to retaining $78k$ (40% pruning), $52k$ (60% pruning), and $13k$ (90% pruning) 3D Gaussians. As can be clearly seen from both the tables, our method consistently and significantly outperforms all baselines across all evaluated pruning rates and metrics by $\sim 80\%$, validating its superiority in achieving high sparsity while maintaining rendering quality. Fig 6 shows the histogram of mean scale values demonstrating our insight empirically that mean scale values of 3D Gaussians correspond-

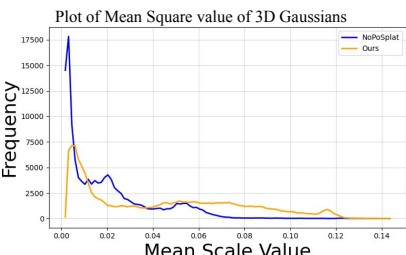

Figure 6: Plot of mean scale values of 3D Gaussians predicted larger than traditional 3D Gaussian representation on same scene to cover low texture regions with fewer primitives.

ing to low texture regions are predicted larger in our method to retain quality of scene; thus, redundant 3D Gaussians can be pruned.

We include detailed evaluations across wider pruning ranges, comparisons with/without fine-tuning across different backbones and pruning strategies are also provided in sec E.2 in supplementary material. Overall, the trend remains the same, highlighting the superiority of the proposed approach. We also include cross domian evaluation on DTU dataset in sec E.3 in appendix section.

**Qualitative Analysis.** We present a qualitative comparison of our method in Figures 4 (left half) - 5. In all these results, we highlight two important issues with the baseline methods:

**(a)** For non-finetuned cases, "black patch" artifacts can be consistently seen, which is indicative of empty spaces within 3D scene where Gaussians were removed without any readjustment. Because of this "black patches" artifacts we see lower metrics across all baselines compared whereas our results shows consistent results without any such artifacts and simultaneosly provides high fidelity.

**(b)** For finetuned cases, results struggle with "blobby Gaussian" artifact. This happens because existing pruning methods completely remove all Gaussians from local region without replacing them. To compensate for the empty space, Gaussians from neighboring areas must expand, which distorts their shape and negatively impacts the quality of overall representation. This effect persists even after further finetuning, as fundamental problem of sub-optimal Gaussian allocation remains.

GaussianTrim3R comprehensively mitigates these issues by directly predicting "optimal" 3D Gaussians for each local region in the scene via SuperClusters, without the need for any post-hoc optimization. We take advantage of classical signal processing properties and efficiently process the scene which provides fine grain control on # 3D Gaussians. We observe similar observations in the ACID dataset as well (Figure 4 right half), even when pruning till $26k$ Gaussians (80% pruning).

| ACID Dataset | 78$k$ Gaussians | | | 13$k$ Gaussians | | |
|---|---|---|---|---|---|---|
| Ablation | PSNR↑ | LPIPS↓ | SSIM↑ | PSNR↑ | LPIPS↓ | SSIM↑ |
| Without Gaussian Adaption | 17.439 | 0.445 | 0.490 | 9.201 | 0.688 | 0.118 |
| With Gaussian Adaption | | | | | | |
| Random Pruning Baseline | 22.530 | 0.305 | **0.653** | 17.496 | 0.392 | 0.544 |
| No Contextual Mask | 22.517 | 0.318 | 0.642 | 17.622 | 0.424 | 0.529 |
| Ours | **22.549** | **0.299** | 0.639 | **18.475** | **0.379** | **0.552** |

Table 3: **Ablation on ACID dataset.** We ablate various components of our method, demonstrating their importance.

We also show effects of progressive allocation for a single scene in Figure 5, where we pick three broad pruning rates ($40\%$, $60\%$ and $80\%$) and compare GaussianTrim3R with the baselines. As can be seen, baselines rapidly degrade the scene quality and completely fail to adapt existing Gaussian attributes in higher sparsity regime. In contrast, GaussianTrim3R optimally re-adjusts 3D Gaussians to adapt to higher pruning rates, consistently maintaining the high-scene quality. We show detailed qualitative comparison results of our method with different backbones and pruning strategies, including the random strategy, in sec E.2.

## 4.2 Ablation Studies

In this section, we discuss the key design choices of our method and present ablation experiments on those in Table 3 to validate their effectiveness.

**Impact of Gaussian Adaption.** To show the necessity of adaptive allocation, we ablate our method by directly subsampling Gaussians from low-texture SuperClusters without training the Gaussian Head. Row "Ours - Adaptive Gaussian Allocation" in the table shows that it plays a crucial role in maintaining scene quality after pruning.

**Random Pruning Baseline.** We ablate on the collective importance of all the modules in our method by providing a simple baseline that trains a feed-forward backbone with random pruning and parameter adaptation. Comparing the row "Random Pruning Baseline" with "Ours" shows that both SuperClusters and Texture-Aware ranking are vital components to ensure high quality rendering of the scene.

**Impact of Contextual Mask.** The contextual mask, which guides the Gaussian Prediction Head by indicating their retained positions, is crucial for attribute adaptation. Result of "No Contextual Mask" row in the table shows that omitting this mask significantly degrades performance, confirming its crucial role in maintaining scene quality.

Collectively, these ablations validate the critical roles of various components of our method in achieving superior performance. We provide additional ablations on (a) choice of $K$ in $K$-means clustering in sec A.1, (b) retention ratio of no. of Gaussians in SuperClusters ($N$) in sec A.2, and (c) the impact of Texture-Aware selection of SuperClusters (we replace Texture-Aware selection to Random SuperCluster selection) in sec A.3 of appendix.

## 5 Conclusion and Future Work

We introduce **GaussianTrim3R**, a novel, feed-forward framework for controllable 3D Gaussian pruning. Our method's core novelty is texture-aware pruning strategy that significantly reduces primitives by targeting low-texture regions. By eliminating the need for expensive per-scene optimization, GaussianTrim3R provides a practical solution for controllable 3D reconstruction, achieving a superior quality-efficiency trade-off even at extreme pruning rates. Our current approach faces a key limitation in scenes with smooth surfaces that exhibit high texture, such as intricate wall patterns. In these cases, texture-based metric can misidentify geometrically simple areas as critical, leading to sub-optimal pruning. Future work will focus on integrating a more nuanced understanding of both texture and geometry to enable more robust pruning decisions, paving the way for truly efficient 3D scene representation.

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

# 6 APPENDIX

# A ADDITIONAL ABLATION

## A.1 CHOICE OF K IN K-MEANS CLUSTERING

Our scene partitioning relies on k-means clustering to form SuperClusters. An ablation study on the ACID dataset (Table 4) reveals that increasing k generally improves reconstruction quality. This is because a higher k yields finer-grained SuperClusters, enabling more precise identification of low-texture regions and, consequently, more judicious placement of larger 3D Gaussians for an optimized trade-off between sparsity and visual fidelity. However, increasing $k$ beyond 300 yields only marginal improvements in quality while significantly increasing GPU memory consumption.

| ACID Dataset | | 78k Gaussians | | | 13k Gaussians | | |
|---|---|---|---|---|---|---|---|
| k value | Memory | PSNR↑ | LPIPS↓ | SSIM↑ | PSNR↑ | LPIPS↓ | SSIM↑ |
| k = 5 | 6.6GB | 22.359 | 0.298 | 0.643 | 18.054 | 0.386 | 0.547 |
| k=10 | 7GB | 22.414 | 0.298 | 0.642 | 18.057 | 0.385 | 0.547 |
| k=50 | 11.12GB | 22.416 | 0.303 | 0.636 | 18.063 | 0.385 | 0.547 |
| k=100 | 16.12GB | 22.488 | 0.301 | 0.638 | 18.066 | 0.385 | 0.547 |
| k=300 | 35.28GB | 22.549 | 0.299 | 0.640 | 18.476 | 0.379 | 0.553 |
| k=400 | 45.04GB | 22.559 | 0.298 | 0.641 | 18.146 | 0.383 | 0.548 |

Table 4: **Trade-off Between $k$ and Resource Usage in Pruning.** Comparison of results at different pruning stages for various $k$ values used in scene partitioning. The second column shows the memory consumed during $k - means$ clustering for each $k$. A higher $k$ generally allows for finer partitioning but also increases computational and memory overhead.

## A.2 DIFFERENT PERCENTAGE OF RETAINED 3D GAUSSIANS

Table 6 presents the results of our method when retaining various percentages of 3D Gaussians within low-texture SuperClusters. We observe that retaining $10\%$ of 3D Gaussians yields the best performance, with this optimal balance becoming particularly evident at high pruning levels (e.g., $13k$ 3D Gaussians). This is because aggressive pruning necessitates a precise balance of sparse, representative Gaussians that can effectively fill vacated regions without creating excessive overlap or redundancy. We notice that with increasing the retained $\%$ of primitives in low texture SuperCluster, we see drop in the novel view metric. This is because increasing the retained $\%$ of primitives leads to redundant Gaussians in same region, which leads to suboptimal quality. This insight is also exploited in previous works like Fan et al. (2023); Fang & Wang (2024); Girish et al. (2024).

## A.3 PRUNING BY CHOOSING SUPERCLUSTERS RANDOMLY:

Another ablation we perform is where remove the texture aware pruning and choose SuperClusters randomly instead of taking texture into account. This way we are pruning the 3D Gaussians withing a SuperCluster but the SuperCluster is choosen randomly. The results for the ablation is present in table 7 for RE10K dataset and in table 8 for ACID dataset. The table shows that texture aware selection of SuperCluster is providing better results in most of the cases compared to random selection because texture based selection for pruning within SuperCluster. Fig 11 shows results for artifacts observed in case we use random SuperCluster for pruning rather than texture based method.

## A.4 ABLATION ON RE10K DATASET

We include extended ablations on RealEstate10K across both moderate and aggressive pruning levels in Tab 5. These results confirm that each module Gaussian Adaptation, Masking, and Structured Pruning plays a distinct and essential role.

Key insights from RE10K ablations:

|  | 78K Gaussians | | | 13K Gaussians | | |
|---|---|---|---|---|---|---|
| Ablation | PSNR↑ | LPIPS↓ | SSIM↑ | PSNR↑ | LPIPS↓ | SSIM↓ |
| Without Gaussian Adaptation | 19.809 | 0.399 | 0.617 | 8.066 | 0.680 | 0.131 |
| Without Mask | 20.055 | 0.339 | 0.656 | 18.090 | 0.389 | 0.625 |
| Random Pruning Baseline | **22.452** | 0.239 | **0.744** | 16.580 | 0.458 | 0.486 |
| Ours | 22.294 | **0.235** | 0.735 | **17.848** | **0.314** | **0.645** |

Table 5: Ablation study on the RE10K dataset across different pruning regimes. The results highlight the contribution of each design component in GaussianTrim3R.

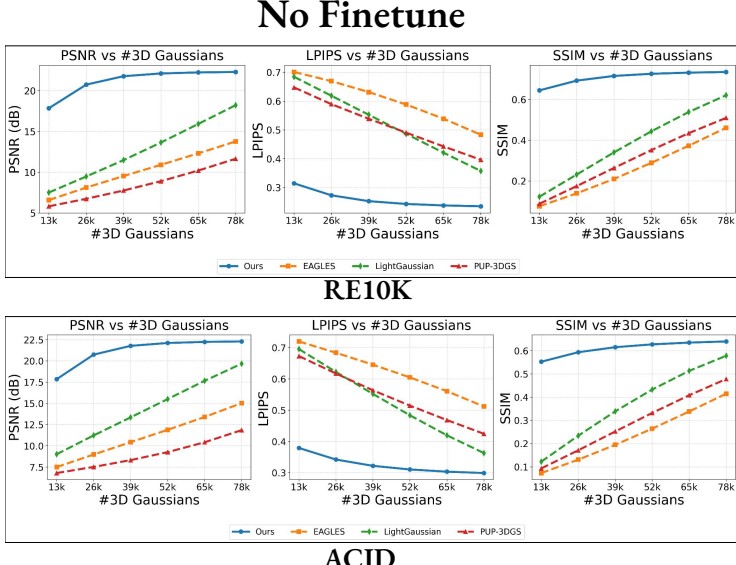

Figure 7: Plot of RE10K and ACID dataset where we compare our results with the relevant baselines without finetunine the remaining 3D Gaussians

- "Without Gaussian Adaptation" feed forward representation collapses under strong pruning ($13K$), presenting that adapting the remaining Gaussians is critical for maintaining scene coverage.

- "Without Mask" performs notably worse than the full model, demonstrating the importance of contextual, texture-aware selection rather than uniform or naïve pruning.

- GaussianTrim3R consistently outperforms all ablations at high compression, validating the necessity of jointly learning masks and adaptive Gaussian updates.

| | **ACID Dataset** | | | | | |
|---|---|---|---|---|---|---|
| | **78k Gaussians** | | | **13k Gaussians** | | |
| Retained % | PSNR↑ | LPIPS↓ | SSIM↑ | PSNR↑ | LPIPS↓ | SSIM↑ |
| 5% | 21.796 | 0.347 | 0.608 | 17.327 | 0.430 | 0.526 |
| 10% | **22.549** | **0.299** | **0.640** | **18.106** | **0.384** | **0.548** |
| 15% | 21.587 | 0.352 | 0.589 | 17.053 | 0.494 | 0.429 |
| 20% | 22.262 | 0.304 | 0.629 | 17.452 | 0.451 | 0.496 |

Table 6: Empirical results on retaining different percetages of 3D Gaussians in low texture Super-Cluster, from the results we conclude that retaining 10% 3D Gaussians yields the best results

## With Finetune

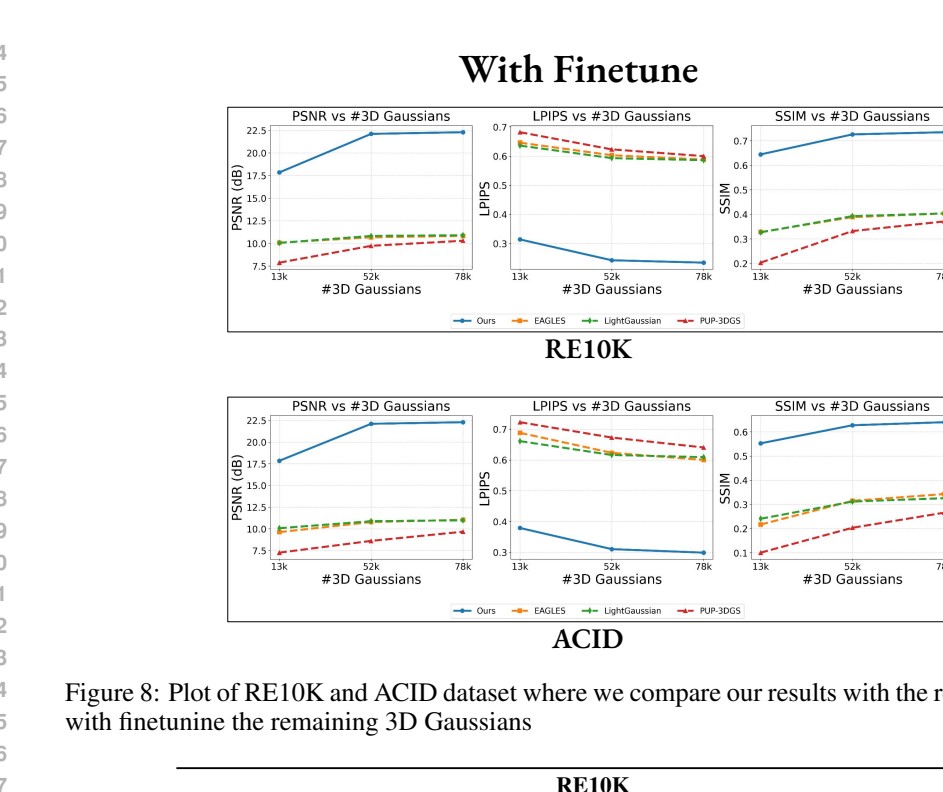

**RE10K**

**ACID**

Figure 8: Plot of RE10K and ACID dataset where we compare our results with the relevant baselines with finetunine the remaining 3D Gaussians

| RE10K | | | | |
|---|---|---|---|---|
| # 3D Gaussians | SuperCluster Selection | PSNR↑ | LPIPS↓ | SSIM↑ |
| 78k | Random | 20.7253 | 0.2602 | 0.7139 |
| | Wavelet (Ours) | **22.2936** | **0.2346** | **0.7352** |
| 65k | Random | 20.2198 | 0.2694 | 0.7051 |
| | Wavelet (Ours) | **22.2385** | **0.2376** | **0.7319** |
| 52k | Random | 19.6852 | 0.2788 | 0.6951 |
| | Wavelet (Ours) | **22.1105** | **0.2427** | **0.7263** |
| 39k | Random | 19.1280 | 0.2884 | 0.6838 |
| | Wavelet (Ours) | **21.7732** | **0.2524** | **0.7157** |
| 26k | Random | 18.5428 | 0.2975 | 0.6713 |
| | Wavelet (Ours) | **20.7403** | **0.2727** | **0.6927** |

Table 7: Ablation results on RE10K dataset to support texture based SuperCluster Selection. Here we chose SuperClusters randomly instead of using texture.

| ACID | | | | |
|---|---|---|---|---|
| # 3D Gaussians | SuperCluster Selection | PSNR↑ | LPIPS↓ | SSIM↑ |
| 78k | Random | 21.3055 | 0.3258 | 0.6227 |
| | Wavelet (Ours) | **22.5490** | **0.2991** | **0.6399** |
| 65k | Random | 20.8880 | 0.3348 | 0.6148 |
| | Wavelet (Ours) | **22.4656** | **0.3034** | **0.6349** |
| 52k | Random | 20.4400 | 0.3437 | 0.6061 |
| | Wavelet (Ours) | **22.3093** | **0.3105** | **0.6274** |
| 39k | Random | 19.9569 | 0.3529 | 0.5964 |
| | Wavelet (Ours) | **21.9617** | **0.3220** | **0.6151** |
| 26k | Random | 19.4199 | 0.3619 | 0.5855 |
| | Wavelet (Ours) | **21.0559** | **0.3422** | **0.5935** |

Table 8: Ablation results on ACID dataset to support wavelet based SuperCluster Selection

| Method | 78K | | | 52K | | | 13K | | |
|--------|-------|-------|-------|-------|-------|-------|-------|-------|-------|
| | PSNR↑ | LPIPS↓ | SSIM↑ | PSNR↑ | LPIPS↓ | SSIM↑ | PSNR↑ | LPIPS↓ | SSIM↑ |
| EAGLES | 18.747 | 0.480 | 0.512 | 18.716 | 0.488 | 0.514 | 16.514 | 0.547 | 0.463 |
| LightGaussian | 19.439 | 0.453 | 0.519 | 19.176 | 0.476 | 0.507 | 14.333 | 0.572 | 0.356 |
| PUP-3DGS | 17.912 | 0.555 | 0.503 | 17.193 | 0.578 | 0.492 | 13.177 | 0.662 | 0.384 |
| Ours | **22.549** | **0.299** | **0.640** | **22.310** | **0.310** | **0.627** | **18.476** | **0.379** | **0.553** |

Table 9: **Results on Additional Baseline on ACID dataset:** As suggested we created additional baselines where the mask is created using the pruning methods (EAGLES, LightGaussians, PUP-3DGS) and used in Gaussian Head which predicts the attributes of retained Gaussians. We consistently outperform the baselines across all pruning strengths by significant margin.

| Method | 78K | | | 52K | | | 13K | | |
|--------|-------|-------|-------|-------|-------|-------|-------|-------|-------|
| | PSNR↑ | LPIPS↓ | SSIM↑ | PSNR↑ | LPIPS↓ | SSIM↑ | PSNR↑ | LPIPS↓ | SSIM↑ |
| EAGLES | 17.828 | 0.412 | 0.555 | 17.726 | 0.421 | 0.551 | 14.437 | 0.514 | 0.435 |
| LightGaussian | 17.964 | 0.395 | 0.545 | 17.599 | 0.416 | 0.531 | 13.004 | 0.534 | 0.394 |
| PUP-3DGS | 15.519 | 0.585 | 0.501 | 15.276 | 0.602 | 0.490 | 11.794 | 0.666 | 0.387 |
| Ours | **22.294** | **0.235** | **0.735** | **22.110** | **0.243** | **0.726** | **17.848** | **0.314** | **0.645** |

Table 10: **Results on Additional Baseline on RE10K dataset:** We've implemented additional baseline as suggested where we create mask using the existing pruning methods and append to Gaussian Head which predicts attributes of retained Gaussians. We consistently outperform the baselines by significant margin across all pruning strengths.

## B    ADDITIONAL BASELINES

As suggested by the reviewers, we implemented additional baselines where pruning masks are first generated using existing pruning pipelines and then concatenated with the Gaussian Head input to predict the attributes of the retained Gaussians. The Gaussian Head for these baselines is trained following the same procedure and loss functions as in our method to ensure fairness.

We evaluate these baselines on both the ACID and RE10K datasets, and the results are provided in Tables 9 and 10. These enhanced baselines achieve stronger results than the post-hoc pruning baselines reported in the main paper, particularly at higher pruning ratios. However, our method still consistently outperforms them across all pruning levels and datasets, primarily due to our texture-aware region selection strategy, which better preserves scene structure under aggressive pruning.

It is also important to note that these additional baselines are not feed-forward: they require (1) predicting the full set of 3D Gaussians, (2) generating a pruning mask, and only then (3) predicting the attributes of the retained Gaussians in a second pass. This makes them inherently two-stage. In contrast, our approach performs pruning within the feed-forward generation pipeline, offering controllable pruning during inference.

## C    INFERENCE LATENCY

To highlight the practical benefits of controllable pruning, we report rendering throughput under different pruning strengths. We adopt the InstantSplat setting for sparse-view reconstruction, where MASt3R provides the initial pixel-aligned points and the 3D Gaussian attributes are subsequently optimized. To ensure a fair comparison, we disable densification so that the number of Gaussians remains fixed to the MASt3R initialization. We used Tanks and Temples dataset Knapitsch et al. (2017) specifically Horse scene for this analysis.

We measure rendering speed by synthesizing 1000 novel views for each pruning stage and reporting the Frames Per Second (FPS). We utilize our pruning technique to prune the Gaussians. Table 11 summarizes the FPS achieved across pruning strengths, and Figure 9 visualizes this trend. As pruning strength increases, the reduced number of active Gaussians consistently improves rendering speed, demonstrating the effectiveness of pruning as a controllable knob for balancing reconstruction quality and runtime efficiency.

| Pruning Strength | 0% | 40% | 50% | 60% | 70% | 80% | 90% |
|---|---|---|---|---|---|---|---|
| **FPS** | 191.4 | 226.7 | 236.3 | 255.0 | 281.4 | 301.3 | 313.8 |
| **PSNR** | 25.744 | 23.932 | 23.859 | 23.692 | 23.573 | 23.530 | 23.974 |

Table 11: **Rendering speed vs. pruning strength**. FPS increases steadily as more Gaussians are pruned while maintaining scene fidelity, demonstrating the utility of pruning as a mechanism for real-time, controllable efficiency during inference. The PSNR was obtained for each pruning strength on Tanks and Temples Horse scene.

| Method | Time (s) | PSNR (13K) |
|---|---|---|
| Baselines (with finetuning) | 6.58 | 9.626 |
| Baselines (without finetuning) | 2.03 | 7.500 |
| Ours | 2.53 | 17.845 |

Table 12: **Inference latency comparison across methods.** Our approach achieves latency similar to the non finetuned baseline while being significantly faster than baselines that require per-scene fine-tuning of retained Gaussians. This demonstrates that GaussianTrim3R preserves fast feed-forward inference while supporting controllable pruning.

We report the end-to-end latency of predicting 3D Gaussians from sparse input views. Table 12 summarizes the runtime across different baselines. Our method exhibits latency comparable to the non-finetuned baseline and is substantially faster than baselines that require per-scene finetuning of the remaining Gaussians. This highlights that GaussianTrim3R retains the feed-forward efficiency of MASt3R while enabling controllable pruning without the overhead of optimization.

# D RESULTS ON INSTANTSPLAT

As suggested by the reviewers, we additionally evaluate our texture-aware pruning strategy on the InstantSplat Fan et al. (2024) framework. We used Tanks and Temples Knapitsch et al. (2017) dataset for our experiment, specifically we used 'Horse' scene from Tanks and Temples. We first use MASt3R to obtain the pixel-aligned point cloud from sparse-view images, construct texture-based SuperClusters, and rank them using our wavelet textureness metric. We prune points according to the desired pruning strength and then train the 3DGS stage for 1000 iterations. Importantly, we disable densification and pruning within InstantSplat to ensure that the Gaussian count remains fixed throughout the optimization pipeline.

We report results for 3, 4, and 10-view input settings and evaluate on novel views (Table 13). Our method consistently achieves higher PSNR, SSIM, and lower LPIPS across all pruning strengths and view configurations. This demonstrates that the proposed texture-aware pruning not only preserves reconstruction quality but also enhances efficiency in multiview sparse-input pipelines without requiring any architectural changes to InstantSplat.

| Pruning Strength | | 40% | | | 60% | | | 80% | | |
|---|---|---|---|---|---|---|---|---|---|---|
| **Input View** | **Method** | **PSNR** | **LPIPS** | **SSIM** | **PSNR** | **LPIPS** | **SSIM** | **PSNR** | **LPIPS** | **SSIM** |
| 3 View | Baseline | 21.565 | 0.217 | 0.745 | 19.320 | 0.321 | 0.662 | 15.482 | 0.479 | 0.508 |
| | Our | 21.536 | 0.215 | 0.762 | 21.464 | 0.261 | 0.746 | 21.484 | 0.276 | 0.747 |
| 4 View | Baseline | 21.815 | 0.201 | 0.735 | 19.673 | 0.310 | 0.681 | 15.564 | 0.469 | 0.510 |
| | Our | 23.932 | 0.176 | 0.820 | 23.692 | 0.215 | 0.804 | 23.530 | 0.236 | 0.799 |
| 10 View | Baseline | 24.754 | 0.141 | 0.852 | 20.655 | 0.249 | 0.746 | 14.827 | 0.416 | 0.558 |
| | Our | 27.649 | 0.113 | 0.896 | 27.310 | 0.138 | 0.886 | 27.178 | 0.154 | 0.882 |

Table 13: **Evaluation of our texture-aware pruning method on InstantSplat across 3-, 4-, and 10-view inputs.** For each pruning strength, we apply our SuperCluster-based pruning before InstantSplat optimization and disable densification to keep the Gaussian count fixed. Across all settings, our method consistently preserves significantly higher reconstruction fidelity (PSNR, SSIM) and lower perceptual error (LPIPS) compared to InstantSplat's native pruning-by-optimization baseline.

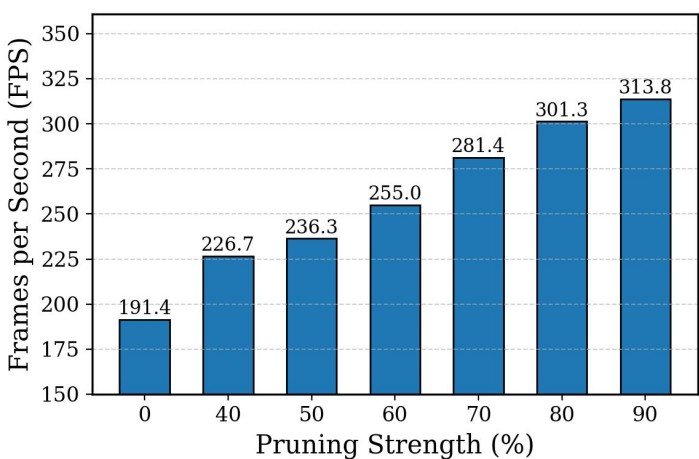

Figure 9: Plot for FPS with Pruning Strength

| Pruning Method | 131K Gaussians | | | 124K Gaussians | | | 118K Gaussians | | | 78k Gaussians | | | 52k Gaussians | | | 13k Gaussians | | |
|---|---|---|---|---|---|---|---|---|---|---|---|---|---|---|---|---|---|---|
| | PSNR↑ | LPIPS↓ | SSIM↑ | PSNR↑ | LPIPS↓ | SSIM↑ | PSNR↑ | LPIPS↓ | SSIM↑ | PSNR↑ | LPIPS↓ | SSIM↑ | PSNR↑ | LPIPS↓ | SSIM↑ | PSNR↑ | LPIPS↓ | SSIM↑ |
| EAGLES | 25.999 | 0.19 | 0.782 | 22.505 | 0.275 | 0.676 | 21.154 | 0.318 | 0.644 | 15.028 | 0.511 | 0.415 | 11.883 | 0.605 | 0.264 | 7.500 | 0.720 | 0.073 |
| LightGaussian | 25.999 | 0.19 | 0.782 | 14.156 | 0.496 | 0.404 | 14.148 | 0.496 | 0.404 | 19.670 | 0.363 | 0.578 | 15.502 | 0.484 | 0.433 | 9.010 | 0.695 | 0.122 |
| PUP-3DGS | 25.999 | 0.19 | 0.782 | 20.981 | 0.268 | 0.666 | 19.274 | 0.289 | 0.649 | 11.845 | 0.424 | 0.478 | 9.257 | 0.514 | 0.332 | 6.797 | 0.673 | 0.093 |
| Ours | 22.843 | 0.272 | 0.662 | 22.815 | 0.273 | 0.661 | 22.787 | 0.274 | 0.659 | 22.294 | 0.299 | 0.640 | 22.110 | 0.310 | 0.627 | 17.845 | 0.379 | 0.553 |

Table 14: **We show results of** $0\%$ **pruning and how it degrades over gradual increase in pruning strength in ACID dataset:** We include the results of $0\%$ pruning as well as nominal pruning at $5\%$ and $10\%$ along with modereate pruning $40\%$ and $60\%$ followed by extreme pruning of $90\%$.

# E ADDITIONAL RESULTS

## E.1 RESULTS ACROSS PRUNING STRENGTHS INCLUDING 0% PRUNING

Tables 14 and 15 report reconstruction quality at 0% pruning together with nominal (5%, 10%), moderate (40%, 60%), and extreme (90%) pruning levels on ACID and RE10K. These results provide a complete view of how pruning impacts both baselines and our method. Fig 10 presents trend of evaluation metric across various pruning strengths from $0\%$ to 90%. **(1) Baseline sensitivity to pruning.** The feed-forward baseline (NoPoSplat) degrades sharply even under minimal pruning: on both datasets, PSNR drops by over 3 dB at only 5% pruning and continues to fall rapidly as pruning increases. This confirms that pixel-aligned Gaussian prediction is highly brittle to Gaussian removal, consistent with the known redundancy of feed-forward 3DGS models. **(2) Stability of our method under light pruning.** In contrast, GaussianTrim3R remains stable across 0–10% pruning, showing less than 0.1 dB variation in PSNR. Although our 0% performance is slightly lower than the dense backbone (since our GS Head is trained to operate across multiple pruning regimes), the model retains scene fidelity even when baselines exhibit early collapse. This highlights the benefit of pruning-aware training and adaptive Gaussian attribute refinement. **(3) Superior robustness at moderate to extreme pruning.** At 40–90% pruning, our method consistently outperforms all baselines across PSNR, SSIM, and LPIPS on both datasets. Even under 90% pruning, GaussianTrim3R preserves coherent structure, while baseline methods fail due to missing geometry and "blobby" ar-

| Pruning Method | 131K Gaussians | | | 124K Gaussians | | | 118K Gaussians | | | 78k Gaussians | | | 52k Gaussians | | | 13k Gaussians | | |
|---|---|---|---|---|---|---|---|---|---|---|---|---|---|---|---|---|---|---|
| | PSNR↑ | LPIPS↓ | SSIM↑ | PSNR↑ | LPIPS↓ | SSIM↑ | PSNR↑ | LPIPS↓ | SSIM↑ | PSNR↑ | LPIPS↓ | SSIM↑ | PSNR↑ | LPIPS↓ | SSIM↑ | PSNR↑ | LPIPS↓ | SSIM↑ |
| EAGLES | 25.056 | 0.161 | 0.839 | 21.886 | 0.218 | 0.751 | 19.914 | 0.265 | 0.711 | 13.769 | 0.483 | 0.461 | 10.908 | 0.588 | 0.289 | 6.591 | 0.701 | 0.076 |
| LightGaussian | 25.056 | 0.161 | 0.839 | 11.552 | 0.498 | 0.406 | 11.552 | 0.498 | 0.406 | 18.218 | 0.357 | 0.621 | 13.642 | 0.486 | 0.444 | 7.513 | 0.684 | 0.123 |
| PUP-3DGS | 25.056 | 0.161 | 0.839 | 18.793 | 0.236 | 0.714 | 17.544 | 0.254 | 0.696 | 11.649 | 0.396 | 0.510 | 8.901 | 0.490 | 0.352 | 5.823 | 0.647 | 0.088 |
| Ours | 22.391 | 0.218 | 0.747 | 22.326 | 0.220 | 0.744 | 22.294 | 0.221 | 0.743 | 22.294 | 0.235 | 0.735 | 22.110 | 0.243 | 0.726 | 17.848 | 0.314 | 0.645 |

Table 15: **We show results of** $0\%$ **pruning and how it degrades over gradual increase in strength in RE10K dataset:** We include the results of $0\%$ pruning as well as nominal pruning at $5\%$ and $10\%$ along with modereate pruning $40\%$ and $60\%$ followed by extreme pruning of $90\%$.

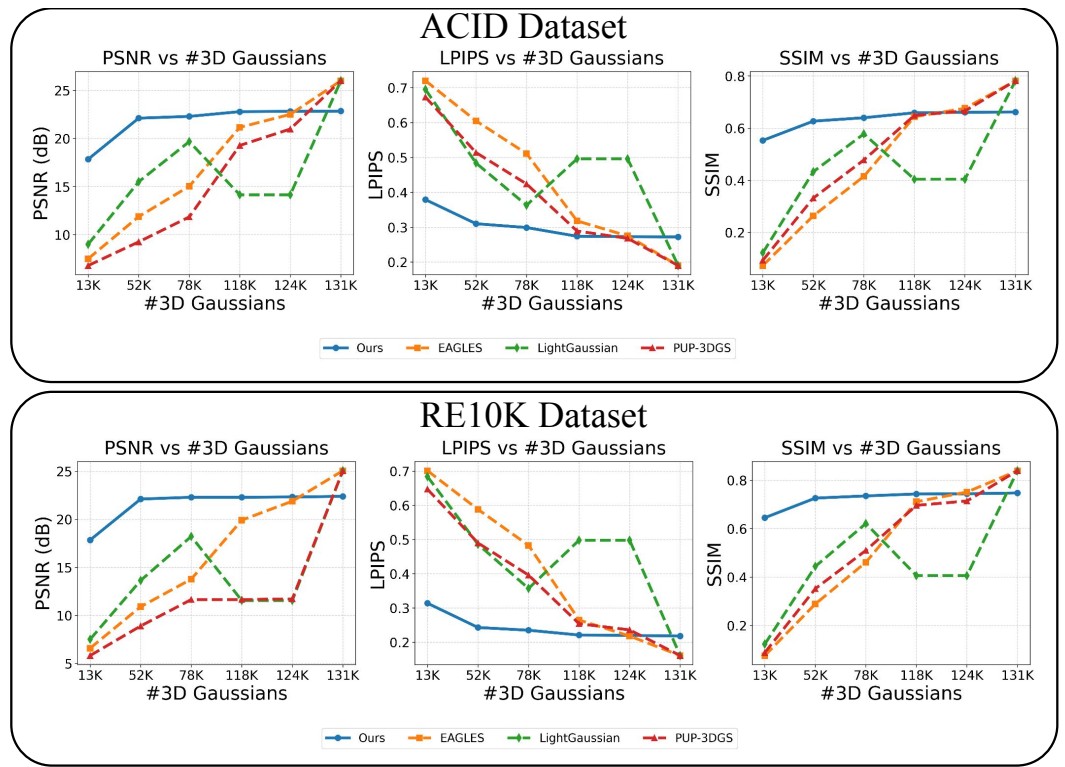

Figure 10: Results of 0% Pruning and gradual increase in pruning strength on both ACID and RE10K dataset

tifacts. The improvements stem from texture-aware clustering, sensitivity-guided pruning, and the GS Head's ability to adapt retained Gaussians to compensate for removed neighbors.

**(4) Use-case positioning.** Our method is intended for scenarios where pruning is needed to meet memory or rendering-speed constraints. When no pruning is required, the base model model can simply be used for full prediction. GaussianTrim3R enables continuous, real-time control over the quality efficiency tradeoff without per-scene optimization, particularly excelling in the moderate-to-severe pruning regime where practical gains are largest.

### E.2 RESULTS ON DIFFERENT BACKBONES

Here we present detailed results of our experiments on RE10K and ACID dataset for different pruning stages on NoPoSplat backbone. Table 16 and table 18 shows results for different pruning stages for ACID and RE10K dataset respectively without finetuning. Whereas table 17 shows results on RE10K dataset after finetuning for pruning at different pruning strategies.

We also include results with different feed-forward backbones with various pruning strategies. Specifically we use pixelSplat and MVSplat backbones on RE10K and ACID dataset with all pruning strategies discussed before (we've also added random 3D Gaussian pruning) along with before and after finetuning results. Table 19 on RE10K dataset and table 20 on ACID dataset presents results of all pruning stages for 2 different feed-forward backbone on 4 different pruning strategies where we don't fine tune the remaining 3D Gaussians after pruning. The table demonstrates that our method achieves better results across most of the pruning stages consistently than any other baselines. Similarly table 22 for ACID dataset and table 21 for RE10K dataset contains results for different feed-forward backbones where we finetune the remaining 3D Gaussians.

We've also included a detailed comparison of each scene with all baselines for a clearer understanding of our method's performance. Fig 13, 14, 15, 16 and 17 shows visual results on ACID dataset for pruning stages of $40\%, 50\%, 60\%, 70\% and 80\%$ of 3D Gaussians where we take random prun-

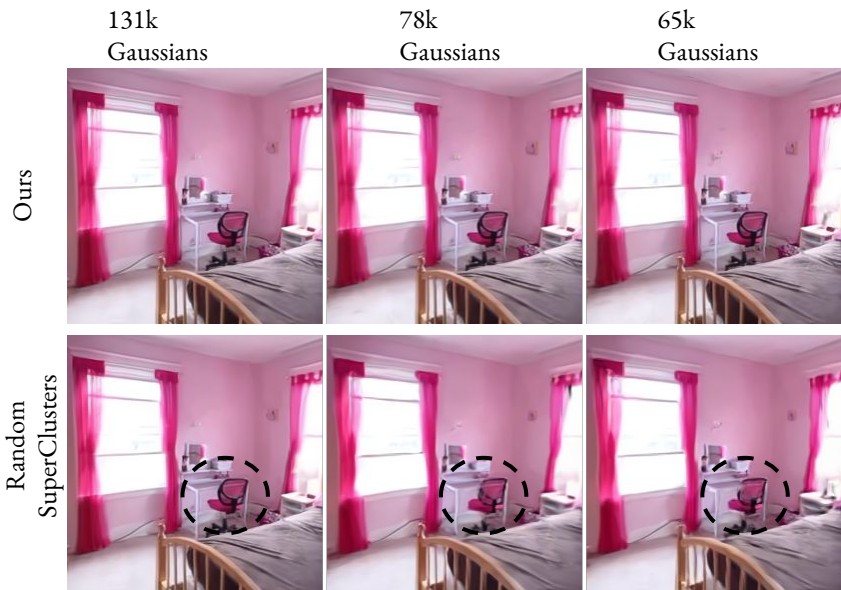

Figure 11: **Results of Ablations on RE10K:** The ablation method begins to lose information around the edges of the chair, particularly in the circular area, while our method retains the details more effectively.

| | ACID Dataset | | | | | | | | | | | |
|---|---|---|---|---|---|---|---|---|---|---|---|---|
| Pruning Method (Without Finetuning) | 78k Gaussians | | | 52k Gaussians | | | 26k Gaussians | | | 13k Gaussians | | |
| | PSNR↑ | LPIPS↓ | SSIM↑ | PSNR↑ | LPIPS↓ | SSIM↑ | PSNR↑ | LPIPS↓ | SSIM↑ | PSNR↑ | LPIPS↓ | SSIM↑ |
| EAGLES | 15.028 | 0.511 | 0.415 | 11.883 | 0.605 | 0.264 | 8.977 | 0.683 | 0.131 | 7.500 | 0.720 | 0.073 |
| LightGaussian | 19.670 | 0.363 | 0.578 | 15.502 | 0.484 | 0.433 | 11.228 | 0.622 | 0.233 | 9.010 | 0.695 | 0.122 |
| PUP-3DGS | 11.845 | 0.424 | 0.478 | 9.257 | 0.514 | 0.332 | 7.516 | 0.617 | 0.171 | 6.797 | 0.673 | 0.093 |
| EfficientGS | 15.189 | 0.446 | 0.468 | 12.316 | 0.524 | 0.376 | 8.886 | 0.653 | 0.196 | 7.414 | 0.708 | 0.102 |
| Ours | **22.294** | **0.2991** | **0.640** | **22.110** | **0.310** | **0.627** | **20.740** | **0.342** | **0.593** | **17.845** | **0.379** | **0.553** |

Table 16: Results on ACID dataset with various existing pruning method without finetuning the remainig using NoPoSplat as backbone

ing strategy on various feed-forward backbone. Similarly fig 18, 19, 20, 21 and 22 shows results on RE10K dataset for different backbone with random pruning of 3D Gaussians. Fig 8 and 7 shows the trends of PSNR, LPIPS and SSIM over different pruning percentages with and without finetuning on RE10K and ACID datasets with and without finetunig 3D Gaussians respectively with NoPoSplat backbone. We can see from the plot that we consistently outperform all the baselines in all pruning scenarios

Fig. 23 presents an example scene evaluated at different 3D Gaussian counts on ACID and RE10K, where we use random pruning on differnet feed-forward backbone. Similarly, Fig. 24 shows the same scene compared against baselines incorporating the LightGaussian pruning method.

E.3    CROSS DOMAIN EVALUATION

We also evaluate the cross domain performance of our model by trainig the model on one particular dataset and evaluating on other. Specifically we trained our model on ACID dataset and evaluated on DTU dataset, similarly we also train on RE10K dataset and evaluated on DTU dataset. Table 23 shows results where we train the baseline on ACID dataset and evaluated on DTU dataset whereas table 24 is trained on RE10K dataset and evaluated on DTU dataset. The baseline that we use is specifically build on NoPoSplat where we use LightGaussian pruning as one baseline and random pruning as another. We use only NoPoSplat as pixelSplat and MVSplat are not trained on large scale diverse dataset as MASt3R used in NoPoSplat. The results shows that our method achieves superior results compared to baselines for the same reason as we discussed in main paper.

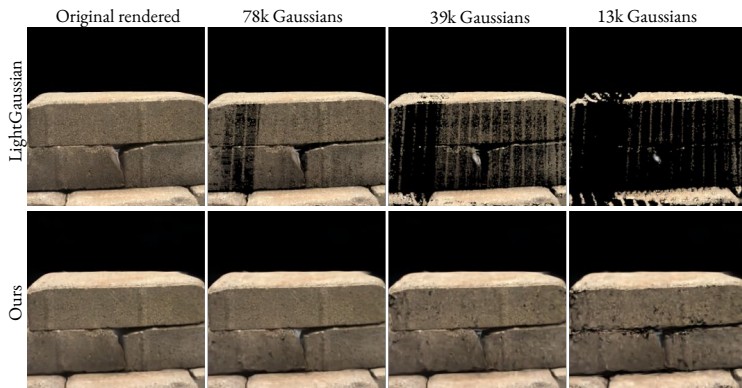

Figure 12: **Cross-Domain Generalization on DTU**: We train our method on the ACID dataset and evaluate its performance on the DTU dataset to test its out-of-distribution generalization capabilities. Our method outperforms baseline approaches in zero-shot pruning, demonstrating its ability to generalize across different datasets without prior training on the target domain.

| **RE10K Dataset** | | | | | | | | | |
|---|---|---|---|---|---|---|---|---|---|
| Pruning Method (With Finetuning) | 78k Gaussians | | | 52k Gaussians | | | 13k Gaussians | | |
| | PSNR↑ | LPIPS↓ | SSIM↑ | PSNR↑ | LPIPS↓ | SSIM↑ | PSNR↑ | LPIPS↓ | SSIM↑ |
| EAGLES | 10.834 | 0.588 | 0.404 | 10.679 | 0.603 | 0.389 | 10.086 | 0.646 | 0.328 |
| LightGaussian | 10.910 | 0.586 | 0.404 | 10.829 | 0.593 | 0.393 | 10.058 | 0.636 | 0.327 |
| PUP-3DGS | 10.298 | 0.600 | 0.371 | 9.737 | 0.623 | 0.332 | 7.878 | 0.682 | 0.203 |
| EfficientGS | 8.742 | 0.658 | 0.262 | 8.841 | 0.655 | 0.266 | 8.604 | 0.671 | 0.252 |
| Ours | **22.294** | **0.235** | **0.735** | **22.110** | **0.243** | **0.726** | **17.848** | **0.314** | **0.645** |

Table 17: Results on RE10K dataset with various existing pruning method with finetuning the remainig Gaussians with NoPoSplat backbone

# F    IMPLEMENTATION DETAILS

Our architecture is based on NopoSplat, which utilizes a Vision Transformer (ViT) as its encoder model. The ViT processes images by dividing them into $16 \times 16$ patches. The weights of the encoder-decoder and point cloud head are initialized from MAST3R, while the Gaussian head is initialized randomly. The embedded camera intrinsics are converted into feature tokens address scale ambiguity and concatenated with the input image tokens as proposed in NopoSplat. We train our model on $256 \times 256$ resolution images and perform all comparisons with baselines at the same resolution for consistency. We trained our model on 1 NVIDIA H100 GPU for 48 GPU hours on each dataset. We use available pretrained checkpoints of the baseline feed-forward models for the evaluation.

| **RE10K Dataset** | | | | | | | | | | | | |
|---|---|---|---|---|---|---|---|---|---|---|---|---|
| Pruning Method (Without Finetuning) | 78k Gaussians | | | 52k Gaussians | | | 26k Gaussians | | | 13k Gaussians | | |
| | PSNR↑ | LPIPS↓ | SSIM↑ | PSNR↑ | LPIPS↓ | SSIM↑ | PSNR↑ | LPIPS↓ | SSIM↑ | PSNR↑ | LPIPS↓ | SSIM↑ |
| EAGLES | 13.769 | 0.483 | 0.461 | 10.908 | 0.588 | 0.289 | 8.118 | 0.669 | 0.139 | 6.591 | 0.701 | 0.076 |
| LightGaussian | 18.218 | 0.357 | 0.621 | 13.642 | 0.486 | 0.444 | 9.476 | 0.618 | 0.231 | 7.513 | 0.684 | 0.123 |
| PUP-3DGS | 11.649 | 0.396 | 0.510 | 8.901 | 0.490 | 0.352 | 6.743 | 0.589 | 0.175 | 5.823 | 0.647 | 0.088 |
| Ours | **22.294** | **0.235** | **0.735** | **22.110** | **0.243** | **0.726** | **20.740** | **0.273** | **0.693** | **17.848** | **0.314** | **0.645** |

Table 18: Results on RE10K dataset with various existing pruning method without finetuning the remainig Gaussians

**RE10K (No Finetune)**

| #3D Gaussians Method | 78k PSNR↑ | LPIPS↓ | SSIM↑ | 65k PSNR↑ | LPIPS↓ | SSIM↑ | 52k PSNR↑ | LPIPS↓ | SSIM↑ | 39k PSNR↑ | LPIPS↓ | SSIM↑ | 26k PSNR↑ | LPIPS↓ | SSIM↑ | 13k PSNR↑ | LPIPS↓ | SSIM↑ |
|---|---|---|---|---|---|---|---|---|---|---|---|---|---|---|---|---|---|---|
| pixelsplat + RandomPruning | 14.67 | 0.477 | 0.528 | 13.22 | 0.515 | 0.466 | 11.70 | 0.554 | 0.398 | 10.10 | 0.594 | 0.323 | 8.43 | 0.637 | 0.240 | 6.70 | 0.686 | 0.143 |
| mvSplat + RandomPruning | 19.27 | 0.444 | 0.567 | 16.99 | 0.501 | 0.472 | 14.65 | 0.556 | 0.378 | 12.28 | 0.611 | 0.286 | 9.88 | 0.664 | 0.195 | 7.43 | 0.713 | 0.104 |
| NoPoSplat + RandomPruning | 19.81 | 0.400 | 0.617 | 18.04 | 0.459 | 0.534 | 15.95 | 0.517 | 0.437 | 13.58 | 0.573 | 0.334 | 10.95 | 0.627 | 0.231 | 8.07 | 0.681 | 0.132 |
| pixelsplat + LightGaussian | 21.32 | 0.349 | 0.720 | 20.58 | 0.383 | 0.694 | 19.83 | 0.413 | 0.670 | 18.68 | 0.450 | 0.637 | 16.23 | 0.517 | 0.570 | 12.79 | 0.601 | 0.456 |
| pixelsplat + EAGLES | **24.750** | **0.193** | **0.840** | **23.142** | 0.244 | **0.805** | 20.518 | 0.313 | **0.753** | 17.718 | 0.399 | 0.621 | 13.576 | 0.492 | 0.468 | 9.793 | 0.584 | 0.288 |
| pixelsplat + PUP-3DGS | 16.208 | 0.382 | 0.619 | 15.411 | 0.394 | 0.610 | 14.163 | 0.417 | 0.587 | 12.679 | 0.452 | 0.543 | 11.005 | 0.502 | 0.469 | 9.026 | 0.578 | 0.343 |
| MVSplat + LightGaussian | 7.71 | 0.693 | 0.132 | 7.33 | 0.703 | 0.114 | 7.02 | 0.711 | 0.100 | 6.69 | 0.718 | 0.087 | 6.19 | 0.730 | 0.068 | 5.77 | 0.739 | 0.051 |
| MVSplat + EAGLES | 7.520 | 0.724 | 0.073 | 11.942 | 0.550 | 0.372 | 10.506 | 0.599 | 0.284 | 9.142 | 0.644 | 0.204 | 7.840 | 0.685 | 0.133 | 6.574 | 0.720 | 0.072 |
| MVSplat + PUP-3DGS | 11.514 | 0.375 | 0.587 | 10.233 | 0.420 | 0.514 | 9.127 | 0.467 | 0.434 | 8.111 | 0.516 | 0.345 | 7.143 | 0.567 | 0.248 | 6.167 | 0.623 | 0.141 |
| NoPoSplat + LightGaussian | 18.20 | 0.361 | 0.619 | 15.92 | 0.424 | 0.536 | 13.67 | 0.488 | 0.442 | 11.52 | 0.553 | 0.339 | 9.50 | 0.619 | 0.230 | 7.52 | 0.685 | 0.121 |
| Ours | 22.29 | 0.235 | 0.735 | 22.24 | **0.238** | 0.732 | **22.11** | **0.243** | 0.726 | **21.77** | **0.252** | **0.716** | **20.74** | **0.273** | **0.693** | **17.85** | **0.314** | **0.645** |

Table 19: Detailed Results on the RE10K dataset comparing different backbone methods across varying pruning techniques. This results are without finetuning the remaining 3D Gaussians.

**ACID (No Finetune)**

| #3D Gaussians Method | 78k PSNR↑ | LPIPS↓ | SSIM↑ | 65k PSNR↑ | LPIPS↓ | SSIM↑ | 52k PSNR↑ | LPIPS↓ | SSIM↑ | 39k PSNR↑ | LPIPS↓ | SSIM↑ | 26k PSNR↑ | LPIPS↓ | SSIM↑ | 13k PSNR↑ | LPIPS↓ | SSIM↑ |
|---|---|---|---|---|---|---|---|---|---|---|---|---|---|---|---|---|---|---|
| pixelSplat + RandomPruning | 15.47 | 0.4968 | 0.4871 | 14.04 | 0.5336 | 0.4260 | 12.54 | 0.5718 | 0.3602 | 10.99 | 0.6126 | 0.2895 | 9.37 | 0.6577 | 0.2124 | 7.69 | 0.7098 | 0.1246 |
| mvSplat + RandomPruning | 18.72 | 0.4799 | 0.4923 | 16.46 | 0.5301 | 0.4101 | 14.28 | 0.5781 | 0.3305 | 12.15 | 0.6243 | 0.2527 | 10.07 | 0.6690 | 0.1752 | 8.02 | 0.7108 | 0.0967 |
| NoPoSplat + RandomPruning | 19.97 | 0.4281 | 0.5364 | 18.20 | 0.4817 | 0.4584 | 16.18 | 0.5329 | 0.3712 | 13.94 | 0.5832 | 0.2823 | 11.49 | 0.6332 | 0.1972 | 8.83 | 0.6850 | 0.1148 |
| pixelSplat + LightGaussian | 23.27 | 0.3789 | 0.6839 | 22.76 | 0.4077 | 0.6596 | 22.34 | 0.4280 | 0.6426 | 21.76 | 0.4514 | 0.6223 | 20.21 | 0.4975 | 0.5820 | 15.95 | 0.5826 | 0.4918 |
| pixelSplat + EAGLES | **25.141** | **0.249** | **0.795** | 22.582 | 0.305 | **0.744** | 19.626 | 0.376 | 0.667 | 16.660 | 0.459 | 0.565 | 13.581 | 0.551 | 0.434 | 10.352 | 0.640 | 0.271 |
| pixelSplat + PUP-3DGS | 19.651 | 0.340 | 0.660 | 17.965 | 0.366 | 0.637 | 13.783 | 0.451 | 0.535 | 11.646 | 0.512 | 0.450 | 11.646 | 0.512 | 0.450 | 9.549 | 0.594 | 0.324 |
| MVSplat + LightGaussian | 8.09 | 0.7199 | 0.0935 | 7.78 | 0.7289 | 0.0812 | 7.54 | 0.7363 | 0.0716 | 7.28 | 0.7441 | 0.0620 | 6.88 | 0.7573 | 0.0481 | 6.53 | 0.7678 | 0.0347 |
| MVSplat + EAGLES | 14.856 | 0.507 | 0.460 | 13.120 | 0.556 | 0.376 | 11.562 | 0.601 | 0.294 | 10.130 | 0.643 | 0.215 | 8.795 | 0.684 | 0.139 | 7.520 | 0.724 | 0.073 |
| MVSplat + PUP-3DGS | 11.429 | 0.395 | 0.563 | 10.258 | 0.441 | 0.488 | 9.323 | 0.488 | 0.410 | 8.536 | 0.538 | 0.328 | 7.808 | 0.590 | 0.239 | 7.050 | 0.648 | 0.141 |
| NoPoSplat + LightGaussian | 19.63 | 0.3638 | 0.5764 | 17.64 | 0.4199 | 0.5116 | 15.49 | 0.4831 | 0.4315 | 13.35 | 0.5509 | 0.3376 | 11.22 | 0.6214 | 0.2330 | 9.00 | 0.6949 | 0.1209 |
| Ours | 22.55 | 0.2991 | 0.6400 | 22.47 | **0.3034** | 0.6349 | 22.31 | **0.3105** | 0.6274 | 21.96 | 0.3220 | 0.6151 | 21.06 | 0.3422 | 0.5935 | 18.48 | 0.3791 | 0.5526 |

Table 20: Detailed Results on the ACID dataset comparing different backbone methods across varying numbers of 3D Gaussians.

**RE10K (With Finetune)**

| #3D Gaussians Method | 78k PSNR | LPIPS | SSIM | 65k PSNR | LPIPS | SSIM | 52k PSNR | LPIPS | SSIM | 39k PSNR | LPIPS | SSIM | 26k PSNR | LPIPS | SSIM | 13k PSNR | LPIPS | SSIM |
|---|---|---|---|---|---|---|---|---|---|---|---|---|---|---|---|---|---|---|
| pixelSplat + EAGLES | 12.155 | 0.703 | 0.104 | 11.911 | 0.694 | 0.104 | 12.991 | 0.670 | 0.131 | 12.399 | 0.618 | 0.188 | 13.034 | 0.612 | 0.240 | 13.210 | 0.605 | 0.265 |
| pixelSplat + PUP-3DGS | 14.311 | 0.610 | 0.282 | 14.309 | 0.607 | 0.287 | 14.151 | 0.607 | 0.290 | 13.623 | 0.612 | 0.286 | 12.475 | 0.623 | 0.271 | 10.669 | 0.646 | 0.234 |
| MVSplat + LightGaussian | 10.937 | 0.702 | 0.113 | 10.799 | 0.702 | 0.113 | 10.403 | 0.702 | 0.126 | 9.983 | 0.717 | 0.119 | 9.054 | 0.710 | 0.109 | 8.035 | 0.720 | 0.090 |
| MVSplat + EAGLES | 11.439 | 0.686 | 0.116 | 11.411 | 0.684 | 0.130 | 11.203 | 0.683 | 0.140 | 10.709 | 0.689 | 0.141 | 9.786 | 0.706 | 0.122 | 8.247 | 0.737 | 0.075 |
| MVSplat + PUP-3DGS | 9.195 | 0.683 | 0.055 | 8.728 | 0.683 | 0.054 | 8.249 | 0.683 | 0.053 | 7.723 | 0.684 | 0.053 | 7.117 | 0.685 | 0.052 | 6.365 | 0.688 | 0.046 |

Table 21: Detailed Results on the RE10K dataset with finetning after pruning comparing different backbone methods across varying pruning techniques.

**ACID (With Finetune)**

| #3D Gaussians Method | 78k PSNR↑ | LPIPS↓ | SSIM↑ | 65k PSNR↑ | LPIPS↓ | SSIM↑ | 52k PSNR↑ | LPIPS↓ | SSIM↑ | 39k PSNR↑ | LPIPS↓ | SSIM↑ | 26k PSNR↑ | LPIPS↓ | SSIM↑ | 13k PSNR↑ | LPIPS↓ | SSIM↑ |
|---|---|---|---|---|---|---|---|---|---|---|---|---|---|---|---|---|---|---|
| pixelSplat + EAGLES | 12.606 | 0.688 | 0.069 | 12.460 | 0.665 | 0.108 | 12.991 | 0.637 | 0.166 | 13.572 | 0.618 | 0.214 | 13.993 | 0.607 | 0.247 | 13.831 | 0.615 | 0.246 |
| pixelSplat + PUP-3DGS | 14.311 | 0.610 | 0.282 | 14.309 | 0.607 | 0.287 | 14.151 | 0.607 | 0.290 | 13.623 | 0.612 | 0.286 | 12.475 | 0.623 | 0.271 | 10.669 | 0.646 | 0.234 |
| MVSplat + LightGaussian | 11.986 | 0.688 | 0.136 | 11.410 | 0.695 | 0.131 | 10.804 | 0.703 | 0.119 | 9.983 | 0.717 | 0.101 | 9.276 | 0.729 | 0.085 | 8.378 | 0.744 | 0.065 |
| MVSplat + EAGLES | 12.495 | 0.677 | 0.122 | 12.423 | 0.677 | 0.135 | 12.154 | 0.679 | 0.144 | 11.600 | 0.687 | 0.142 | 10.616 | 0.705 | 0.118 | 9.029 | 0.739 | 0.069 |
| MVSplat + PUP-3DGS | 9.262 | 0.680 | 0.046 | 8.725 | 0.683 | 0.041 | 8.273 | 0.687 | 0.037 | 7.894 | 0.692 | 0.036 | 7.524 | 0.697 | 0.037 | 7.061 | 0.706 | 0.035 |

Table 22: Detailed Results on the ACID dataset with finetning after pruning comparing different backbone methods across varying pruning techniques.

**Cross Domain on DTU from ACID data trained model (No finetune)**

| #3D Gaussians | 78k PSNR↑ | LPIPS↓ | SSIM↑ | 65k PSNR↑ | LPIPS↓ | SSIM↑ | 52k PSNR↑ | LPIPS↓ | SSIM↑ | 39k PSNR↑ | LPIPS↓ | SSIM↑ | 26k PSNR↑ | LPIPS↓ | SSIM↑ | 13k PSNR↑ | LPIPS↓ | SSIM↑ |
|---|---|---|---|---|---|---|---|---|---|---|---|---|---|---|---|---|---|---|
| Original Noposplat Model | 14.8391 | 0.4442 | 0.3583 | 14.0862 | 0.4786 | 0.3244 | 13.1471 | 0.5138 | 0.2858 | 11.9843 | 0.5504 | 0.2423 | 10.9908 | 0.5877 | 0.1918 | 8.8578 | 0.6341 | 0.1291 |
| Nopo + Light Gaussian | 14.8479 | 0.399 | 0.3717 | 13.9737 | 0.4401 | 0.3304 | 12.8484 | 0.4854 | 0.2782 | 11.5911 | 0.5362 | 0.2179 | 10.2454 | 0.5867 | 0.154 | 8.7326 | 0.6463 | 0.0891 |
| OURS | 15.23 | 0.4244 | 0.427 | 15.17 | 0.4297 | 0.4218 | 15.12 | 0.4368 | 0.4169 | 15.01 | 0.446 | 0.4113 | 14.76 | 0.46 | 0.4034 | 13.98 | 0.4793 | 0.3906 |

Table 23: **Out-of-distribution performance comparison: Model trained on ACID and evaluated on DTU**

**Cross Domain on DTU from RE10K data trained model (No finetune)**

| #3D Gaussians | 78k PSNR↑ | LPIPS↓ | SSIM↑ | 65k PSNR↑ | LPIPS↓ | SSIM↑ | 52k PSNR↑ | LPIPS↓ | SSIM↑ | 39k PSNR↑ | LPIPS↓ | SSIM↑ | 26k PSNR↑ | LPIPS↓ | SSIM↑ | 13k PSNR↑ | LPIPS↓ | SSIM↑ |
|---|---|---|---|---|---|---|---|---|---|---|---|---|---|---|---|---|---|---|
| Original Noposplat Model | 16.1808 | 0.4053 | 0.5160 | 15.2241 | 0.4453 | 0.4678 | 14.0450 | 0.4857 | 0.4127 | 12.6249 | 0.5258 | 0.3489 | 10.9735 | 0.5674 | 0.2739 | 9.0026 | 0.6146 | 0.1789 |
| Nopo + Light Gaussian | 16.2601 | 0.3547 | 0.5033 | 15.0991 | 0.3981 | 0.4482 | 13.4953 | 0.4522 | 0.3791 | 11.9285 | 0.5114 | 0.2983 | 10.0433 | 0.5693 | 0.2082 | 8.8015 | 0.6339 | 0.1165 |
| OURS | 16.1900 | 0.4073 | 0.4746 | 16.1400 | 0.4153 | 0.4693 | 16.0200 | 0.4252 | 0.4606 | 15.8100 | 0.4368 | 0.4513 | 15.3800 | 0.4519 | 0.4389 | 14.2900 | 0.4764 | 0.4232 |

Table 24: Out-of-distribution performance comparison: Model trained on RE10K and evaluated on DTU

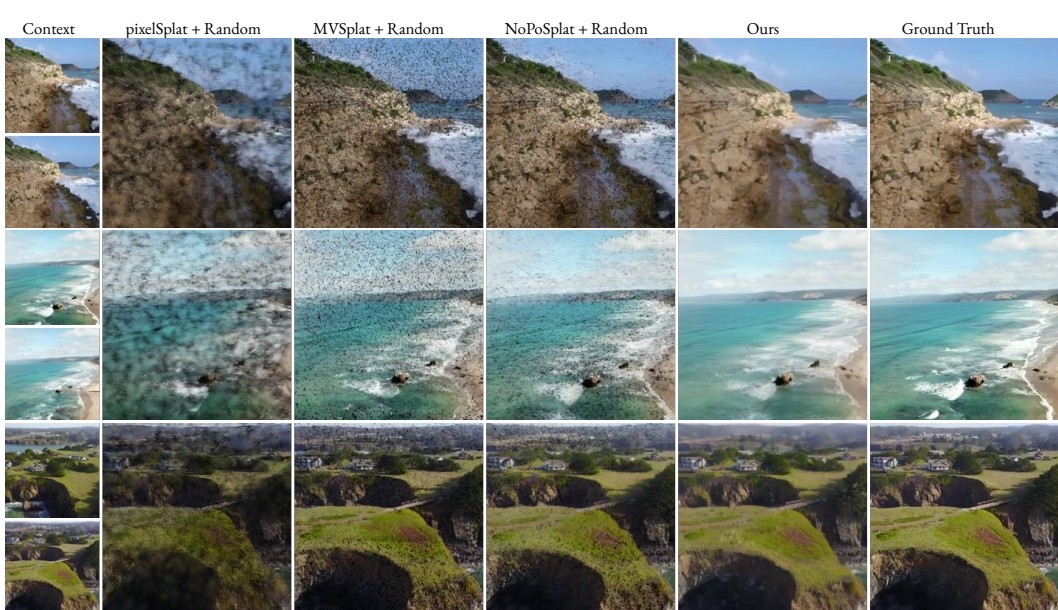

Figure 13: Detailed Results comparison with 78K 3D Gaussians on ACID dataset

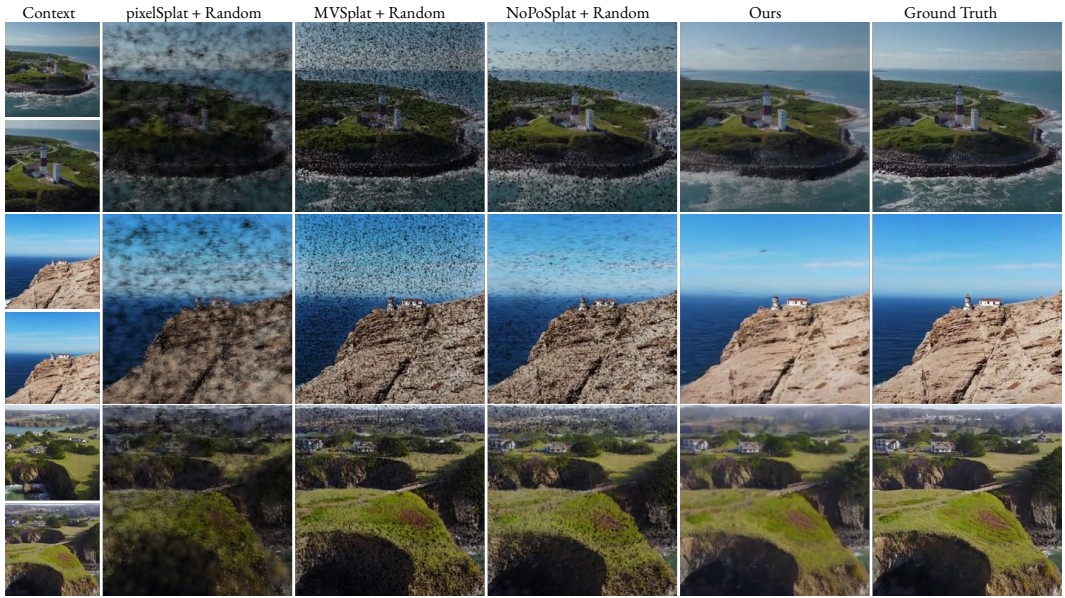

Figure 14: Detailed Results comparison with 65K 3D Gaussians on ACID dataset

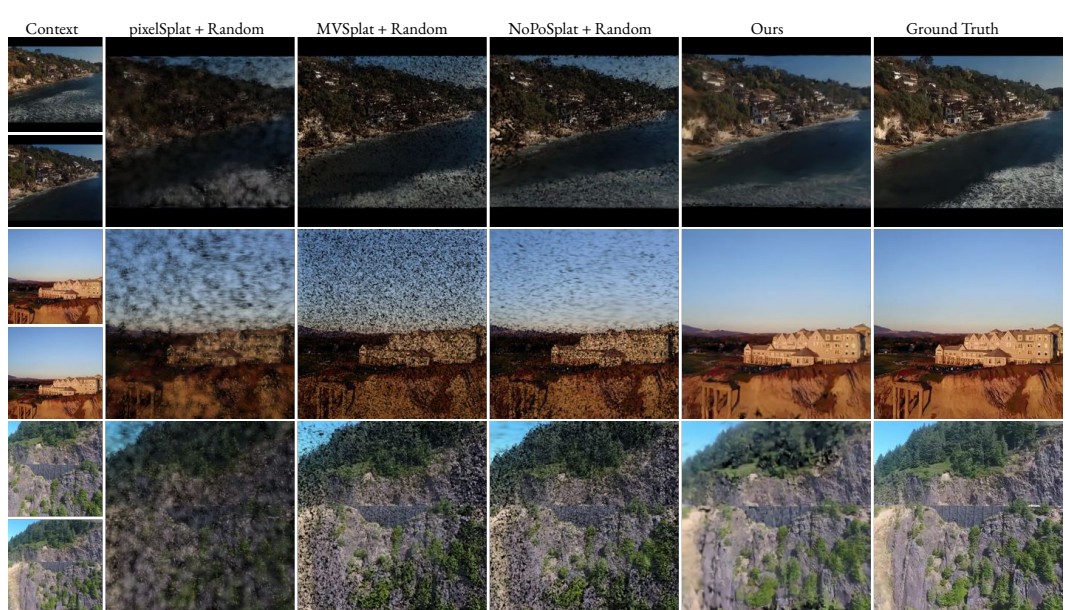

Figure 15: Detailed Results comparison with 52K 3D Gaussians on ACID dataset

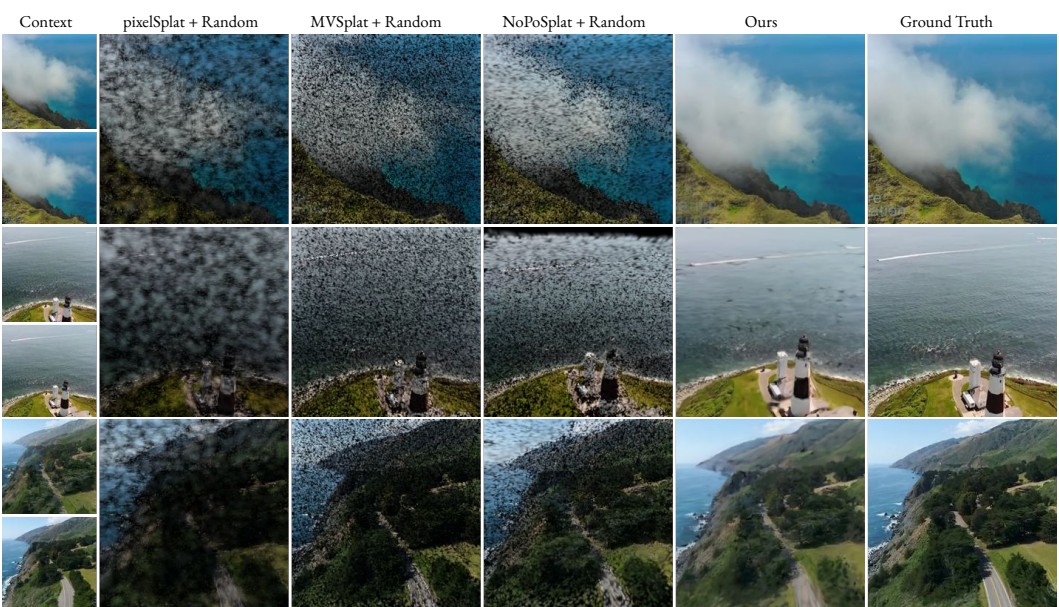

Figure 16: Detailed Results comparison with 39K 3D Gaussians on ACID dataset

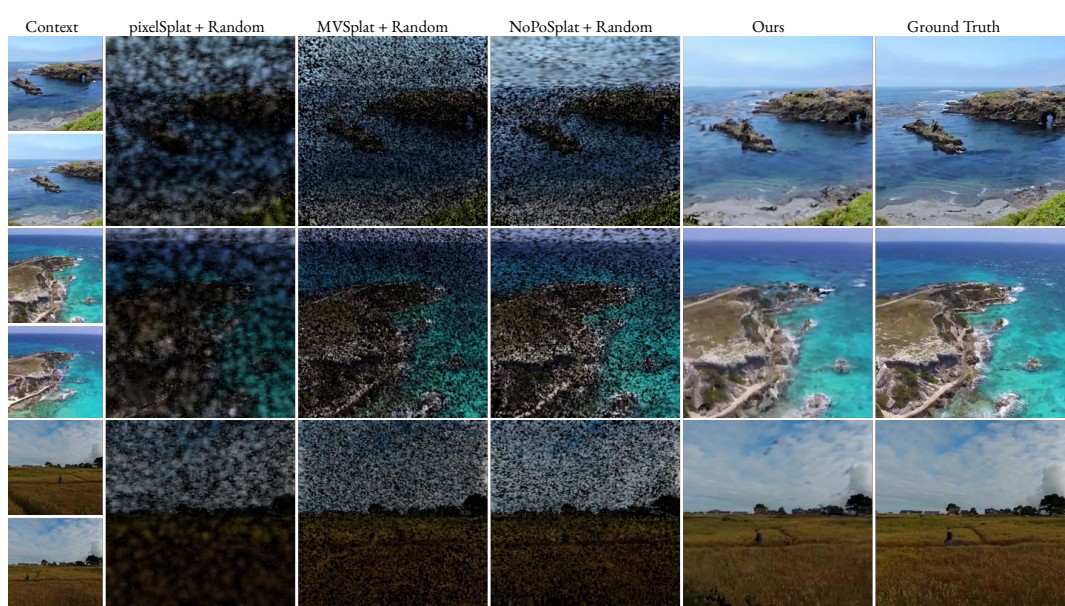

Figure 17: Detailed Results comparison with 26K 3D Gaussians on ACID dataset

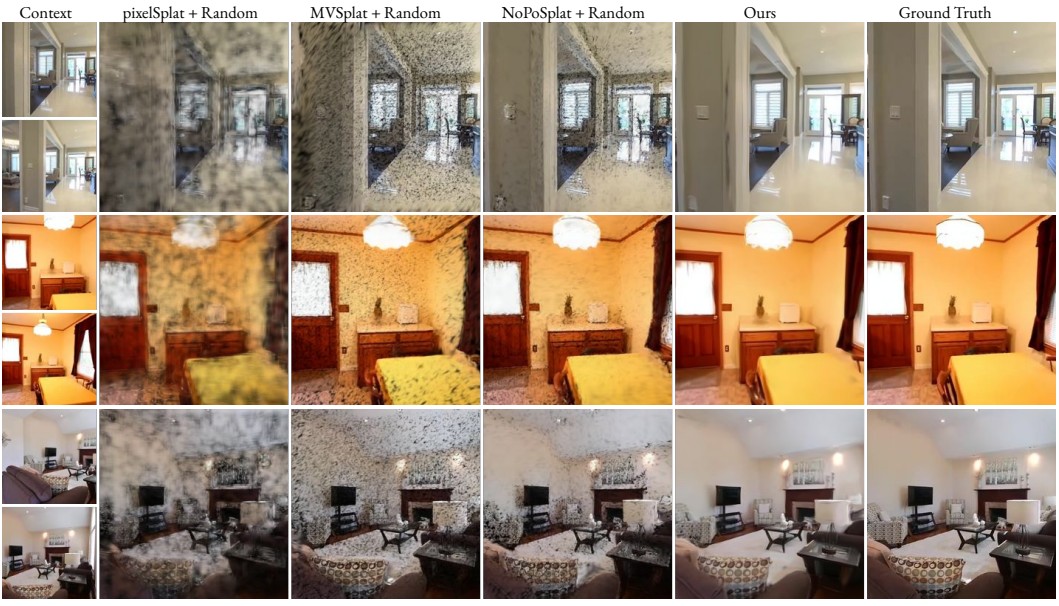

Figure 18: Detailed Results comparison with 78K 3D Gaussians on RE10K dataset

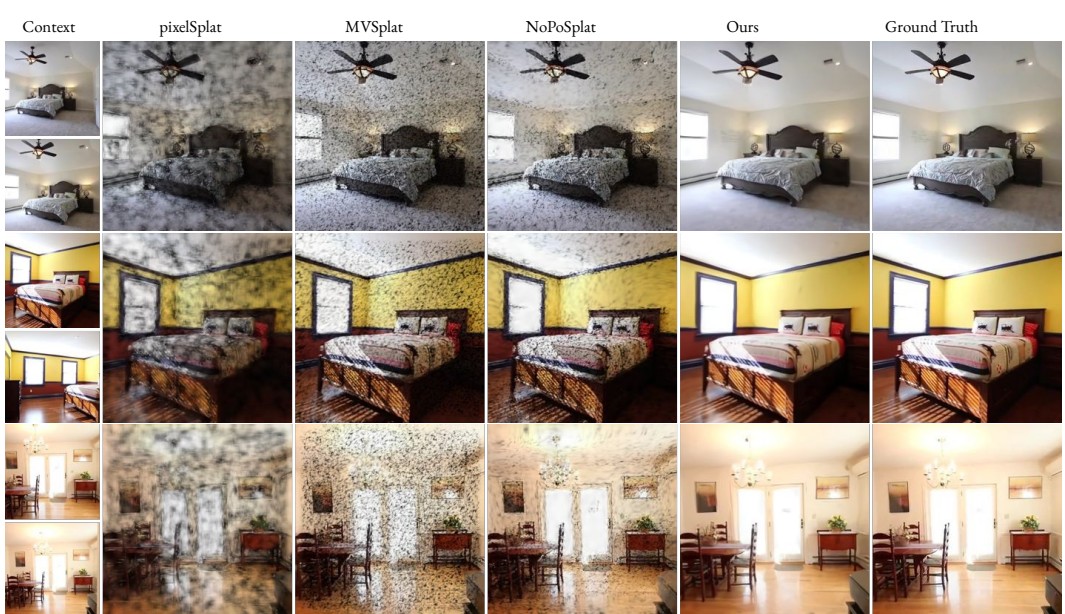

Figure 19: Detailed Results comparison with 65K 3D Gaussians on RE10K dataset

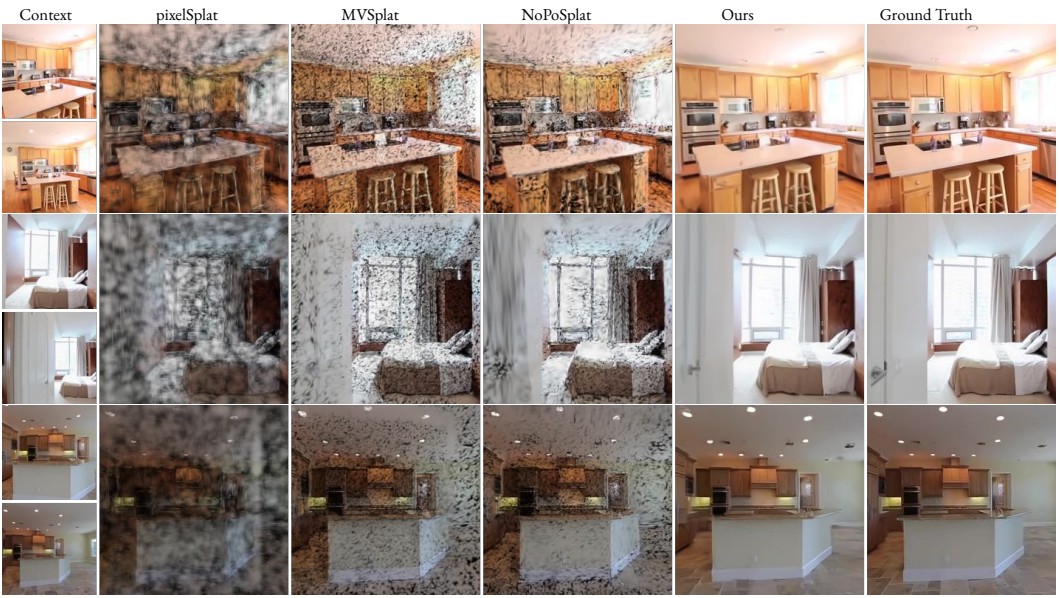

Figure 20: Detailed Results comparison with 52K 3D Gaussians on RE10K dataset

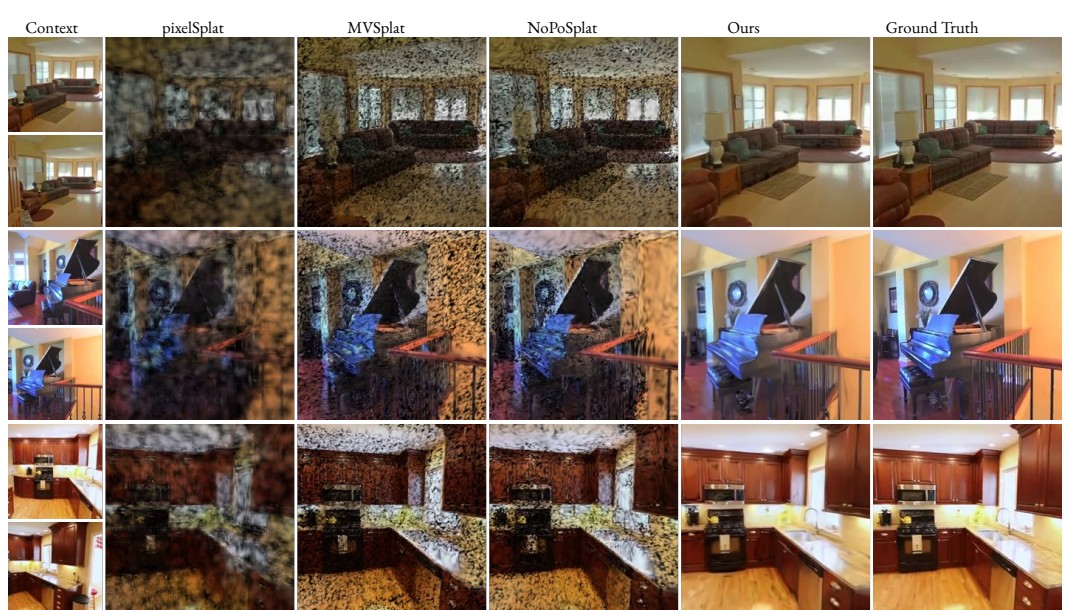

Figure 21: Detailed Results comparison with 39K 3D Gaussians on RE10K dataset

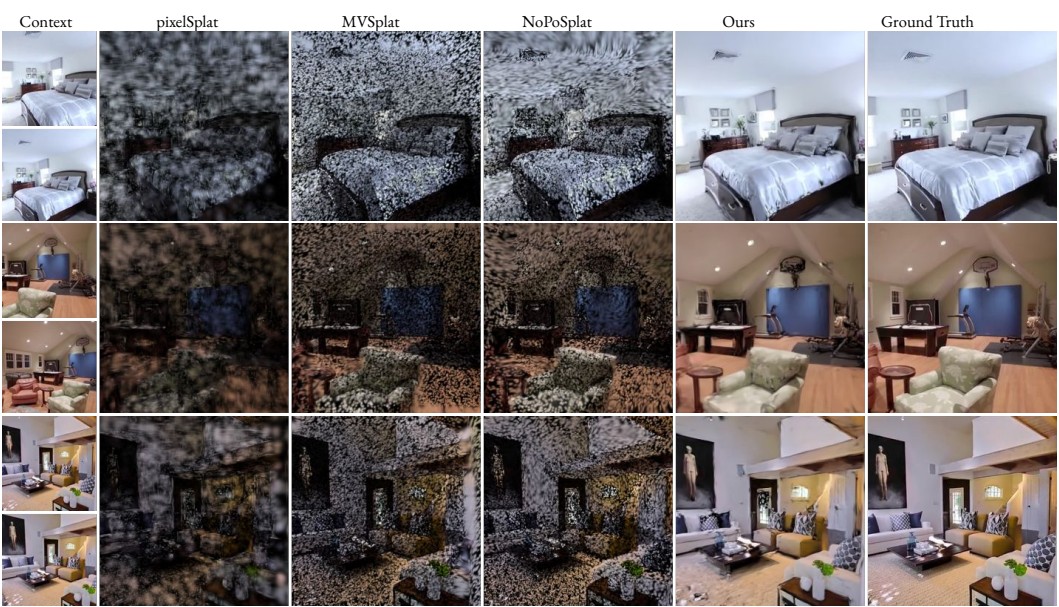

Figure 22: Detailed Results comparison with 26K 3D Gaussians on RE10K dataset

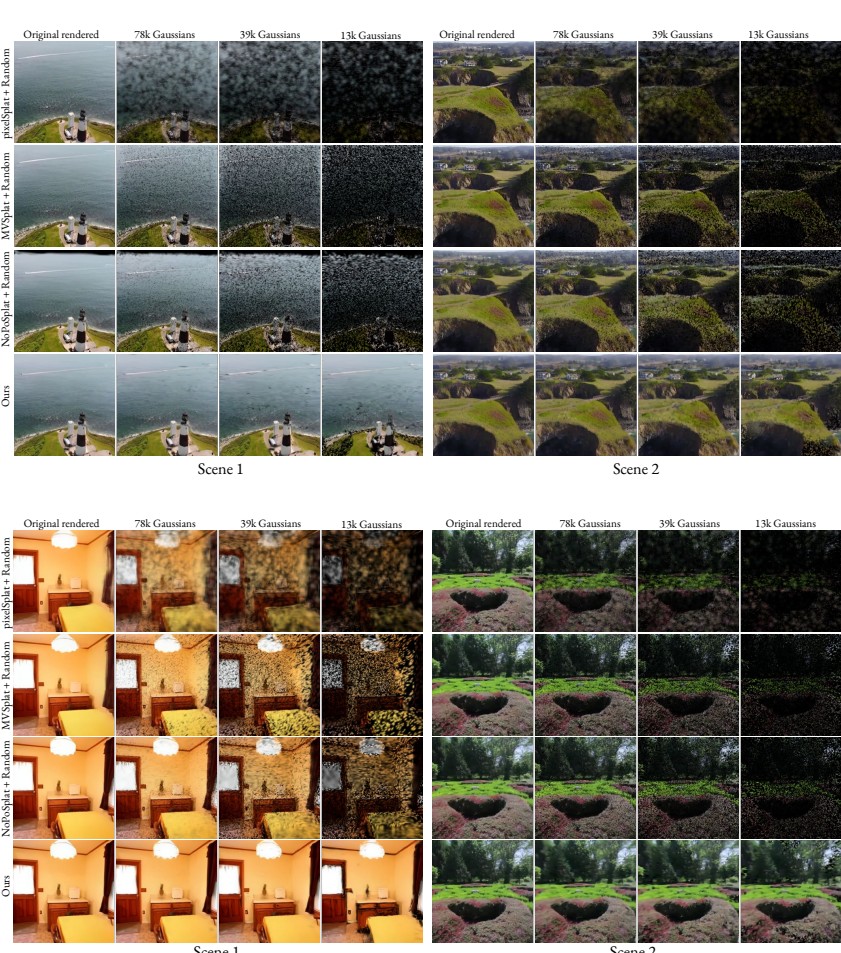

Figure 23: Detailed Comparison of baselines results for ACID (top) and RE10K (bottom) datasets. We perform random prunning on top of the 3D Gaussians obtained from other baselines.

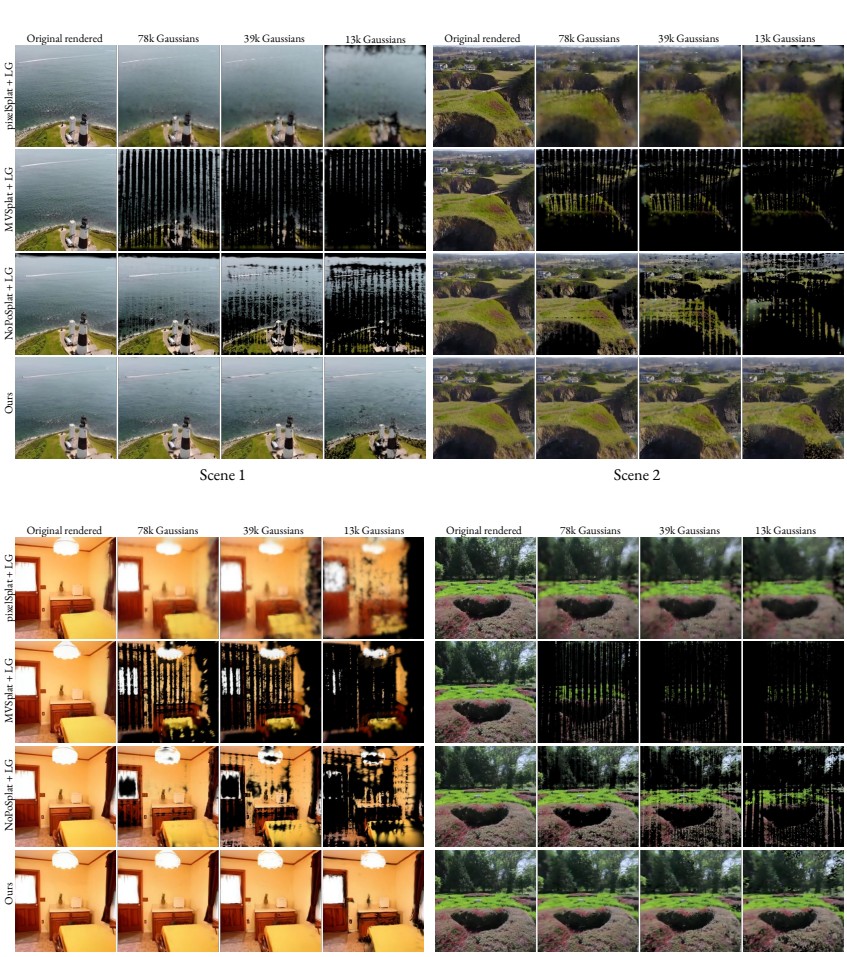

Figure 24: Detailed Comparison of baselines results for ACID (top) and RE10K (bottom) datasets. We perform LightGaussian prunning on top of the 3D Gaussians obtained from other baselines.

