# OpenReview forum: "GaussianTrim3R: Controllable 3D Gaussians Pruning for Feedforward models"
_ICLR.cc/2026/Conference — Submitted to ICLR 2026_

### Official Review · Reviewer_RPAH · 2025-10-23

**Soundness:** 3
**Presentation:** 3
**Contribution:** 3
**Rating:** 4
**Confidence:** 4

**Summary:**

This paper proposes GaussianTrim3R, a feed-forward framework for pruning 3D Gaussian Splatting (3DGS) representations under a limited Gaussian budget. The authors argue that current feed-forward models predict a fixed number of Gaussians, leading to redundancy.

**Strengths:**

1. Addresses a timely and relevant problem: efficient and controllable pruning of 3D Gaussian Splatting for real-time rendering and feed-forward models.

2. Clearly recognizes the redundancy issue in current feed-forward Gaussian prediction frameworks such as MASt3R and DUSt3R.

3. Shows a clear problem motivation and structured pipeline, even though the method itself is heuristic.

**Weaknesses:**

1. 2D texture → 3D pruning mismatch. Textureness is computed on image views via DWT and then averaged per SuperCluster. There is no principled treatment of visibility, occlusions, or multi-view consistency. A 2D texture proxy can be high where geometry is flat but textured (e.g., wallpaper) or low where geometry is complex but uniformly colored; both cases can cause harmful pruning. The paper acknowledges this failure mode but does not quantify it or provide mitigation.

2. Efficiency and “real-time” are not demonstrated. The paper claims real-time, controllable inference but provides no FPS, latency, GPU memory, or throughput numbers under different budgets.

3. Ablations are narrow; component necessity is unclear. The ablation table focuses on a single dataset and a couple of pruning points.

**Questions:**

1. How exactly is the “adaptive Gaussian expansion” implemented? Is it learned or deterministic?

2. What is the exact runtime and memory cost reduction versus MASt3R or DUSt3R backbones?

3. How does your method behave on scenes with high-frequency textures but simple geometry

---

> ### Author Response · Authors · 2025-11-21
>
> We thank the reviewer for their constructive feedback.
>
> # Q1: No treatment on visibility, occlusions or consistency
>
> We clarify that in feed-forward pipelines (e.g., DUSt3R, MAST3R) the predicted 3D points are pixel-aligned to the input images. Thus, the position of each Gaussian primitive is guaranteed to be visible in at least one input view. This property reduces the need for an explicit visibility or occlusion model during pruning. Further, our pruning strategy is applied after the prediction of Gaussian centres and the SuperClusters are formed in 3D space. Hence, if they are visible in at least one view and at least one pixel, they will be considered in our pruning pipeline.
>
> # Q2: Failure mode and how to mitigate
>
> As acknowledged in the manuscript (line 482-483), using 2D textureness as a proxy for 3D geometric complexity can introduce failure modes in scenarios like highly textured surfaces or uniformly colored but geometrically complex regions.
> Importantly, regions with strong 2D texture naturally receive higher textureness scores, making their SuperClusters less likely to be pruned except at extremely high pruning ratios. This behavior can be seen in Fig. 5 of the main paper, where even at 80% pruning on ACID, our method maintains scene fidelity in high-texture areas, whereas baselines fail noticeably.
>  We agree that handling these rare corner cases more explicitly for example, by incorporating multi-view visibility cues or geometric priors would further improve robustness. We plan to explore such extensions in future work.
>
> # Q3: Need FPS, GPU Memory, and Latency
>
> **Providing FPS and latency measurements:**
>  Thank you for raising this important point. We now report full inference-time statistics across pruning levels in section C of the appendix as well as in Table 11 and 12 and Figure 9 in the updated manuscript. As shown below, GaussianTrim3R achieves real-time inference, and the FPS increases consistently as pruning becomes more aggressive, while maintaining scene fidelity making it highly useful in feed forward setting. Even moderate pruning yields meaningful speed-ups, and heavy pruning boosts FPS by ~64% over the unpruned case. This validates that controllable pruning is an effective and practical way to accelerate feed-forward 3DGS inference while preserving reconstruction quality. This trend directly reflects our design objective: pruning reduces the number of rendered Gaussians, which proportionally decreases rendering cost. Importantly, this speed-up requires no additional optimization, and users can adjust pruning strength at test time to achieve a continuous quality speed trade-off based on their application requirements.
>
> ### FPS
> | Pruning Strength | 0%    | 40%   | 50%   | 60%   | 70%   | 80%   |
> |------------------|--------|--------|--------|--------|--------|--------|
> | **FPS**          | 191.4 | 226.7 | 236.3 | 255.0 | 281.4 | 301.3 |
>
>
>
> We have reported inference time for our method along with baselines in both cases (with finetuning and without finetuning) in Table 12 of the updated manuscript. We observe that our method takes a similar time to that of "baseline without finetuning", but we are substantially better than the baselines where remaining Gaussians are finetuned after pruning. Moreover, the "baseline without finetuning" shows poor reconstruction quality.
> ### Inference Time (seconds)
> | Method                          | Time (s) | PSNR (13K) |
> |---------------------------------|----------|-------------|
> | Baselines (with finetuning)     | 6.58     | 9.626       |
> | Baselines (without finetuning)  | 2.03     | 7.500       |
> | **Ours**                        | 2.53 | 17.845  |
>
>
> # Q4: Additional Ablations on RealEstate10K
>
>  We’ve added ablations to RealEstate10K across in section A.4 of the appendix in updated manuscript. These results highlight the complementary roles of the mask module and the adaptive GS Head as well as our pruning method. The results clearly show that all components contribute meaningfully to the pipeline providing flexible representation.
>
> # Q5: Clarifying the Implementation of Adaptive Gaussian Expansion
>
> In our framework, adaptive Gaussian expansion is a learned mechanism, not a deterministic post-processing step. During training, the GS Head observes pruned regions across varying pruning ratios and is optimized (via the loss in Eq. 3, line 302) to adjust the attributes of the retained Gaussians such as scale, covariance, opacity, and orientation so that they naturally expand or adapt to cover regions where Gaussians were removed. The GS Head is trained with the pruning masks, it learns to reallocate representational capacity spatially. When low-texture clusters are aggressively pruned, the GS Head automatically compensates by enlarging or adjusting the surviving Gaussians to preserve scene fidelity and avoid holes. This is what enables our method to remain stable even at extreme compression levels.

---

> > ### Author Response · Authors · 2025-11-21
> >
> > # Q6: Runtime and Memory Cost
> >
> > **Clarifying runtime behavior:**
> > Since our approach builds directly on the MASt3R backbone, our inference-time backbone latency matches MASt3R exactly. The additional DPT-based Gaussian Head introduces a small overhead, placing our total inference time slightly above the NoPoSplat baseline (~50 ms)  while still maintaining real-time capability. Importantly, our method does not require any per-scene optimization and supports controllable pruning directly at inference time.
> >
> > **Clarifying memory usage:**
> >  Our memory footprint matches NoPoSplat, since both methods use MASt3R + a lightweight attribute head. During inference, NoPoSplat uses ~7 GB of GPU memory, and our method operates within the same range.
> >
> > **Runtime Efficiency**
> >
> > We provide assessment of runtime efficiency by comparing our inference latency against both finetuned and non-finetuned baselines. As shown in the table below, our method achieves comparable latency to the non-finetuned baseline, while being substantially faster than baselines that require per-scene finetuning. Importantly, unlike baselines, our method delivers significantly higher reconstruction quality at similar latency.
> > ### Latency (seconds)
> > | Method                          | Time (s) | PSNR (13K) |
> > |---------------------------------|----------|-------------|
> > | Baselines (with finetuning)     | 6.58     | 9.626       |
> > | Baselines (without finetuning)  | 2.03     | 7.500       |
> > | **Ours**                        | **2.53** | **17.845**  |

---

> > > ### Author Response · Authors · 2025-11-25
> > > **Gentle Reminder**
> > >
> > > Dear reviewer RPAH
> > >
> > > As the discussion window is approaching its final week, we wanted to gently follow up and draw your attention to our rebuttal and the accompanying revisions (also summarized at the **“Brief Summary of Paper Revisions”** ). We have carefully addressed each of your comments, including handling visiblity/occlusion, failure mode mitigation, latency analysis, additional ablations and memory cost.
> > >
> > > Thank you for your valuable feedback and for helping improve the paper.
> > >
> > > Best Regards,
> > >
> > > Authors

---

> > > > ### Comment · Reviewer_RPAH · 2025-11-25
> > > > **Respond to Authors**
> > > >
> > > > Thank you for your effort. I have raised the mark.

---

> > > > > ### Author Response · Authors · 2025-11-27
> > > > >
> > > > > Thank you for your careful review and for updating your score. We appreciate your time and consideration.

---

### Official Review · Reviewer_KvJ4 · 2025-10-29

**Soundness:** 3
**Presentation:** 3
**Contribution:** 2
**Rating:** 6
**Confidence:** 4

**Summary:**

The authors propose GaussianTrim3R, a feedforward 3DGS method that can adaptively change the output Gaussian numbers based input masks and targeted budget. GaussianTrim3R first obtains the point cloud from input pair images and perform clustering + frequency analysis. These information come together to rank the texture complexity of point cloud clusters; then GaussianTrim3R select from the clusters with lowest texture to start masking away features, such that the GS head can learn to produce fewer but larger Gaussians at these regions. Such a process is done progressively until certain threshold of Gaussian budget are satisfied. Overall results indicate that this approach is better than random pruning at very high compression rate (10%).

**Strengths:**

The topic of feed forward Gaussian compression is timely, especially when there are a lot of input images. The current scheme of generating Gaussian per-pixel will not be sustainable.

The authors propose to train a mask-aware GS head such that the generated GS can be of larger shapes, which seems reasonable to me.

The overall performance at high compression rate seems to show that pruning over low texture regions lead to differential performance, which makes sense intuitively.

The writing is easy to understand for the most part.

Despite my issues with the baseline comparisons, I am willing to give borderline accept given the simple but straightforward innovation in training feedforward GS models, which hasn't been explored before. I would love if the authors can confirm if e.g., this method can be made to work under e.g., 6-12 views.

**Weaknesses:**

While feed forward Gaussian compression is an important topic, my sense is that Gaussian count only becomes a real problem with a lot of input images. E.g., given two views, representing the scene with 200k Gaussians can be readily supported by normal computers. As such, I think it's worthwhile to see if this work can be applied in scenarios beyond two views.

This also has ramification to comparisons. The baselines that GaussianTrim3R compare with are all designed for multi-view reconstruction, not two-view reconstruction. Particularly, if the original gaussians are not maintained and are pruned instead, it is very hard for these methods to recover the 3D scene, which is conventionally known. As such, these baselines are not very useful - more useful baselines may be e.g., InstantSplat or SPARS3R, which work on sparse view reconstruction. E.g., for InstantSplat, the setting is that existing initialization will not be pruned, and it maybe useful to see the effectiveness of these methods if the initialization can be pruned.

The ablation shows that, random pruning/no contextual mask lead to similar performance at 40% pruning, indicating that there is a lot of redundancy in feedforward GS methods, though the 0% pruning GS metric is not posted. Differentiable results begin to emerge at 10%, which begs the question of whether this method only works when the compression is very severe, which is relatively niche.

**Questions:**

Are all experiments with post-inference finetuning/optimization done with only two view constraint?

It would be very helpful if authors can list the original metrics at 0% pruning so that reviewers understand the full context of pruning.

---

> ### Author Response · Authors · 2025-11-21
>
> We thank the reviewer for their constructive feedback.
> # Q1: Extension to beyond 2 View
> We clarify that our Gaussian pruning strategy for feed-forward models is generalizable to multi-view pipelines :
> - Sequential pipelines such as CUT3R[1] that accumulates estimated 3D points into a common coordinate frame for dense reconstruction across sequential multi-views.
> - Multi-view pipelines such as MV-DUSt3R+[2] take multi-view images as input and predict pixel-aligned Gaussian primitives from a Gaussian head.
>
> Our pruning strategy can be integrated into these pipelines and would result in substantial memory reduction as the number of views increases. Please refer to section C of the appendix in the updated manuscript where we’ve shown gain in FPS as pruning strength increases. We’ve also shown our results on multi-view input on Tanks and Temples dataset in section D of the appendix. We will explore the multi-view extensions as part of future work.
>
> [1] Wang, Qianqian, et al. "Continuous 3d perception model with persistent state." Proceedings of the Computer Vision and Pattern Recognition Conference. 2025.
>
> [2] Tang, Zhenggang, et al. "Mv-dust3r+: Single-stage scene reconstruction from sparse views in 2 seconds." Proceedings of the Computer Vision and Pattern Recognition Conference. 2025.
>
> # Q2: Applying Our Pruning Strategy to InstantSplat
>
> Thank you for this insightful suggestion. We experimented with integrating our wavelet-based pruning into InstantSplat by disabling densification so that the number of input Gaussians remains fixed and consistent with MASt3R’s pixel-aligned initialization. We prune MASt3R points using our method prior to InstantSplat training. The results are included in section D of the Appendix in updated manuscript for your reference.
>
> We observe that the performance drop is minimal even at high pruning ratios, and still maintains strong fidelity while benefiting from faster rendering due to fewer primitives. This demonstrates that our pruning strategy is compatible with multi-view pipelines and provides practical benefits beyond the feed-forward setting.
>
> | Input View | Method   | **40% PSNR** | **40% LPIPS** | **40% SSIM** | **60% PSNR** | **60% LPIPS** | **60% SSIM** | **80% PSNR** | **80% LPIPS** | **80% SSIM** |
> |------------|-----------|--------------|----------------|---------------|---------------|----------------|---------------|---------------|----------------|---------------|
> | **3 View** | Baseline | **21.565**       | 0.217          | 0.743         | 19.320        | 0.321          | 0.662         | 15.482        | 0.479          | 0.508         |
> |            | Ours     | 21.536       | **0.215**          | **0.762**         | **21.464**        | **0.261**          | **0.746**         | **21.484**        | **0.276**          | **0.747**         |
> | **4 View** | Baseline | 21.818       | 0.201          | 0.739         | 19.673        | 0.310          | 0.681         | 15.564        | 0.469          | 0.510         |
> |            | Ours     | **23.932**       | **0.176**          | **0.820**         | **23.692**        | **0.215**          | **0.804**         | **23.530**        | **0.236**         | **0.799**         |
> | **10 View**| Baseline | 24.754       | 0.141          | 0.852         | 20.655        | 0.249          | 0.746         | 14.827        | 0.416          | 0.558         |
> |            | Ours     | **27.649**       | **0.113**          | **0.896**         | **27.310**        | **0.138**          | **0.886**         | **27.178**        | **0.154**          | **0.882**         |
>
>
> # Q3: Behaviour Under Moderate vs. Extreme Pruning
> We agree that feed-forward GS models contain considerable redundancy, which is why random pruning can appear competitive at moderate levels (e.g., 40%). However, as shown in our results, this redundancy disappears rapidly as pruning becomes more aggressive. When the pruning ratio increases to 80 to 90%, random or non-contextual pruning collapses quickly (**Appendix section A.4, and Table 5**), while GaussianTrim3R maintains high PSNR/SSIM and stable rendering quality due to its texture-aware pruning and adaptive Gaussian allocation. This regime is increasingly relevant in practical systems e.g., recent works such as SpeedySplat [1] (CVPR 2025) explicitly target high compression for fast rendering.
> Importantly, GaussianTrim3R is not limited to extreme pruning. A key contribution of our method is continuous, user-controllable pruning at inference without optimization or retraining. Even in cases where moderate pruning shows only modest differences due to redundancy, our method allows users to dynamically adjust the quality speed trade-off depending on device constraints (edge hardware, mobile XR) or application needs.
>
> [1] Hanson, Alex, et al. "Speedy-splat: Fast 3d gaussian splatting with sparse pixels and sparse primitives." Proceedings of the Computer Vision and Pattern Recognition Conference. 2025.

---

> > ### Author Response · Authors · 2025-11-21
> >
> > # Q4: Experiments Beyond 2 Views & Request for 0%-Pruning Metrics
> >
> > ### **Clarifying the experimental setup:**
> > Thank you for raising this point. All feed-forward experiments, including post-inference finetuning/optimization, use a two-view input constraint, consistent with the standard MASt3R/DUSt3R sparse-view setting. However, novel-view rendering is evaluated on target views (4 views) that differ from the input pair, ensuring that pruning quality is assessed in a generalizable multi-view regime.
> >
> > ### **Experiments beyond two views:**
> >
> > To address the reviewer’s concern, we additionally report results for 3-view and 4-view input settings using InstantSplat as the reconstruction backbone. Detailed results are mentioned in section D and Table 13 of the appendix in the updated manuscript. These experiments verify that our pruning strategy remains robust across higher-view regimes as well, further strengthening the generality of our method.
> >
> > | Input View | Method   | **40% PSNR** | **40% LPIPS** | **40% SSIM** | **60% PSNR** | **60% LPIPS** | **60% SSIM** | **80% PSNR** | **80% LPIPS** | **80% SSIM** |
> > |------------|-----------|--------------|----------------|---------------|---------------|----------------|---------------|---------------|----------------|---------------|
> > | **3 View** | Baseline | **21.565**       | 0.217          | 0.743         | 19.320        | 0.321          | 0.662         | 15.482        | 0.479          | 0.508         |
> > |            | Ours     | 21.536       | **0.215**          | **0.762**         | **21.464**        | **0.261**          | **0.746**         | **21.484**        | **0.276**          | **0.747**         |
> > | **4 View** | Baseline | 21.818       | 0.201          | 0.739         | 19.673        | 0.310          | 0.681         | 15.564        | 0.469          | 0.510         |
> > |            | Ours     | **23.932**       | **0.176**          | **0.820**         | **23.692**        | **0.215**          | **0.804**         | **23.530**        | **0.236**          | **0.799**         |
> > | **10 View**| Baseline | 24.754       | 0.141          | 0.852         | 20.655        | 0.249          | 0.746         | 14.827        | 0.416          | 0.558         |
> > |            | Ours     | **27.649**       | **0.113**          | **0.896**         | **27.310**        | **0.138**          | **0.886**         | **27.178**        | **0.154**          | **0.882**         |
> >
> >
> > ### **Providing the requested 0%-pruning metrics:**
> >
> > As suggested, we now include complete 0%-pruning results for both baselines in section E.1 of the appendix of the updated manuscript. These results present the full reconstruction performance prior to pruning.
> >
> > | Pruning Method | **131K** PSNR ↑ | **131K** LPIPS ↓ | **131K** SSIM ↑ | **124K** PSNR ↑ | **124K** LPIPS ↓ | **124K** SSIM ↑ | **118K** PSNR ↑ | **118K** LPIPS ↓ | **118K** SSIM ↑ |
> > |----------------|---------------------------|-------------------|------------------|----------------------------|-------------------|------------------|----------------------------|-------------------|------------------|
> > | EAGLES         | 25.999                    | 0.19              | 0.782            | 22.505                     | 0.275             | 0.676            | 21.154                     | 0.318             | 0.644            |
> > | LightGaussian  | 25.999                    | 0.19              | 0.782            | 14.156                     | 0.496             | 0.404            | 14.148                     | 0.496             | 0.404            |
> > | PUP-3DGS       | 25.999                    | 0.19              | 0.782            | 20.981                     | 0.268             | 0.666            | 19.274                     | 0.289             | 0.649            |
> > | **Ours**       | 22.843                    | 0.272             | 0.662            | 22.815                     | 0.273             | 0.661            | 22.787                     | 0.274             | 0.659            |
> >
> >
> > ### **Interpreting the 0% performance:**
> > As shown below, the baseline models exhibit a noticeable fidelity drop even at nominal 5% and 10% pruning, whereas GaussianTrim3R maintains highly stable performance across 0–10% pruning. The slight reduction at 0% in our method is expected: unlike the backbone (optimized solely for dense Gaussian prediction), our GS Head is trained to operate across a range of pruning strengths, including very sparse ones. This encourages adaptive representational capacity rather than overfitting to dense settings, enabling robust generalization under realistic and resource-constrained pruning budgets.

---

> > > ### Author Response · Authors · 2025-11-25
> > > **Gentle Reminder**
> > >
> > > Dear Reviewer KvJ4,
> > >
> > > We hope you are doing well. With the discussion period approaching its final week, we kindly invite you to take a moment to review our detailed rebuttal and follow-up clarifications. We have addressed all raised concerns thoroughly including methodological questions, extension beyond 2-view, 0% pruning metrics and InstantSplat experiments and updated the manuscript accordingly.
> > >
> > > If any aspect remains unclear or if you would like further results or analysis, we would be very happy to provide them promptly. Your feedback has been extremely helpful, and we truly appreciate your time and effort in reviewing our work.
> > >
> > > Thank you once again for the constructive discussion we look forward to any further comments you may have.
> > >
> > > Warm regards,
> > >
> > > Authors

---

### Official Review · Reviewer_qQcZ · 2025-10-31

**Soundness:** 4
**Presentation:** 4
**Contribution:** 3
**Rating:** 6
**Confidence:** 3

**Summary:**

This paper proposes a feed-forward 3D Gaussian Splatting (3DGS) framework with a primitive-number control algorithm. The method dynamically regulates the number of Gaussian primitives during training and achieves real-time optimization. Specifically, the approach introduces SuperClusters to group the 3D Gaussian primitives and employs a Discrete Wavelet Transform to assess texture complexity, generating masks that guide where and how aggressively to prune the Gaussians. Experimental results demonstrate that the method achieves superior performance compared with baseline approaches.

**Strengths:**

- Pruning is integrated into the 3D Gaussian generation process rather than applied post-training, enabling joint optimization and leading to improved performance.
- The method can accurately identify over-fitted regions and prune them precisely.
- The proposed approach achieves superior quantization results and visual quality compared to baseline methods under the same pruning ratio. Notably, at high pruning ratios, baseline methods tend to collapse, whereas the proposed method maintains strong reconstruction quality.

**Weaknesses:**

- The hyperparameters 𝐾 (number of clusters) and 𝑁 (pruning ratio per patch) play an important role in the proposed method. However, their values appear to be chosen empirically. It would be beneficial to explore adaptive or data-driven strategies for determining these parameters, which could improve robustness and reduce manual tuning efforts.

**Questions:**

I would like clarification regarding the training and inference pipeline. Are the center heads and Gaussian-splat heads trained on a large dataset during the training stage, and then kept frozen during inference? During inference, do we simply follow the framework shown in Figure 3 to produce the 3D Gaussians? Additionally, how many iterations are required to generate the 3D Gaussians at inference time—only a single forward pass, or multiple iterations? Out of curiosity, how much time needed for the inference?

---

> ### Author Response · Authors · 2025-11-21
>
> We thank the reviewer for their constructive feedback
>
> # Q1: Hyperparameter Selection for K (Clusters) and N (Pruning Ratio)
>
> We clarify that the rationale behind selecting both K and N is already detailed in Appendix A.1 of the updated manuscript, but we summarize the key insights here for completeness.
>
> ## Choice of K (number of SuperClusters).
> The value of K = 300 reflects a balance between texture sensitivity and computational feasibility. Increasing K creates smaller and homogeneous SuperClusters which is beneficial for highly textured regions. However, increasing K also introduces extra computational overhead and can lead to over-fragmentation of the data. In our experiments (Table 4, Section A.1 of Appendix), values above 300, such as K = 400, produce clusters that are too small to maintain local coherence, harming reconstruction quality while also exceeding typical GPU memory limits during training. Thus, K = 300 provides the results between memory and SuperCluster size.
>
> ## Choice of N (per-cluster retention ratio).
>  We set the Gaussian retention ratio to 10% based on extensive ablations (Table A.2). Retaining fewer Gaussians (<10%) causes under-coverage and sparse artifacts, particularly in textured regions, while retaining more (>10%) introduces redundancy in smooth areas, reducing pruning effectiveness. A 10% retention ratio preserves essential scene coverage while allowing adaptive expansion of retained Gaussians to compensate for removed ones, yielding an effective balance between efficiency and reconstruction fidelity.
>
> We thank reviewer for suggesting  data-driven strategies for determining these parameters, and we will incorporate this in future work
>
>
> # Q2: Frozen Models During Inference
>
> Both heads are fully trained on the dataset and kept frozen during inference. The inference stage consists of a single forward pass through the frozen network. The pruning strength (budget) is supplied only as a control variable to adjust the mask during inference; no parameters are updated at test time. This design ensures feed-forward behavior and enables smooth control over pruning strengths without any optimization or finetuning.
>
> # Q3: Number of Iterations Required at Inference Time
>
> We confirm that inference strictly follows the pipeline illustrated in Figure 3. The entire scene reconstruction requires only a **single forward pass**, with no iterative refinement or optimization. This is a key property of GaussianTrim3R and distinguishes it from optimization-based 3DGS methods.
>
> # Q4: Total Inference Time
> We have included the details about inference latency in Section C of the Appendix, along with Table 11 and Figure 9 of the updated manuscript (highlighted in blue for easier identification), which shows that the FPS consistently increases as the pruning strength increases while maintaining the reconstruction quality.
> Additionally We have also reported inference time for our method along with baselines in both cases (with finetuning and without finetuning) in Table 12. We observe that our method takes a similar time to that of "baseline without finetuning", but we are substantially better than the baselines where remaining Gaussians are finetuned after pruning. Moreover, the "baseline without finetuning" shows poor reconstruction quality.
> ### Inference Time (seconds)
> | Method                          | Time (s) | PSNR (13K) |
> |---------------------------------|----------|-------------|
> | Baselines (with finetuning)     | 6.58     | 9.626       |
> | Baselines (without finetuning)  | 2.03     | 7.500       |
> | **Ours**                        | **2.53** | **17.845**  |
>
> ### FPS
> | Pruning Strength | 0%    | 40%   | 50%   | 60%   | 70%   | 80%   |
> |------------------|--------|--------|--------|--------|--------|--------|
> | **FPS**          | 191.4 | 226.7 | 236.3 | 255.0 | 281.4 | 301.3 |

---

> > ### Author Response · Authors · 2025-11-25
> > **Gentle Reminder**
> >
> > Dear Reviewer qQcZ
> >
> > With the discussion phase concluding soon, we kindly invite you to have a look at our rebuttal and updates. We have carefully addressed all the concerns you raised, including:
> >
> > - Clarifying the methodological questions,
> > - Clarifying questions related to selection of K (SuperClusters) and N(Pruning Ratio)
> > - Reporting inference latency, FPS, and memory behavior
> >
> > All revisions in the appendix are highlighted in blue for quick reference.
> > If you have any further questions or would like additional clarification, we would be very happy to continue the discussion.
> >
> > Thank you again for your time and support during the review process.
> >
> > Warm regards,
> > Authors

---

### Official Review · Reviewer_D4Jq · 2025-11-01

**Soundness:** 3
**Presentation:** 2
**Contribution:** 4
**Rating:** 6
**Confidence:** 5

**Summary:**

This paper presents a novel method for generating lightweight, high-fidelity Gaussian scenes from sparse views. Prior zero-shot methods generate pixel-aligned 3D Gaussians, which often leads to duplicated Gaussians in simple-textured areas. To address this issue, given initial points, the authors identify point clusters (K=300) and, starting from the low-textured clusters ranked by Equation 2, iteratively prune 100–N% of the Gaussians in each cluster until reaching the target Gaussian budget (Z=13k, 52k, 78k). For the experiments, since this problem is newly defined by the authors, they constructed baselines by combining existing zero-shot generation and pruning methods. Compared with these baselines, their method demonstrates significant improvements across various pruning ratios. The ablation study further justifies the necessity of the proposed components.

**Strengths:**

This paper defines a new problem: how to generate lightweight Gaussian scenes from sparse images. It neatly addresses the problem by identifying texture-less regions, which require fewer Gaussians for representation, generating a mask, and then generating Gaussians with consideration of the mask. Although some representations still need refinement, the paper is generally clear and easy to follow.

**Weaknesses:**

Please see the Questions section for the major concerns.

Presentation issues:
* In line 089, doesnt -> doesn't
* In Figure 4, the authors need to annotate which columns correspond to 40% and 80% of the Gaussians.
* In Figure 4, it would be a good idea to juxtapose a non-finetuned image corresponding to the same scene as the finetuned image to illustrate the "blobby Gaussian" artifact of the baseline.
* In Figure 4, I suggest representing the distributions using blue and yellow line curves without filling for NoPoSplat and your method. Currently, the authors fill the overlapping area with brown, which is understandable only when I check color composition palette.
* In Tables 1 and 2, I suggest reporting the performance of the baselines and GaussianTrim3R without pruning as a reference to show how much the performance of pruned version degrades from their full capacity.

**Questions:**

* My biggest question is whether the baselines are truly designed fairly. The gap of 8 dB seems phenomenal, but upon some reflection, it makes sense because the baselines apply pruning methods after generating Gaussians, whereas GaussianTrim3R’s GS Head directly generates hole-free scenes by leveraging a mask. I suggest a baseline that first generates pixel-wise initial Gaussians using the Mast3r backbone with some trainable Initial GS Head, then applies pruning methods to these initial Gaussians. Based on the pruning scores, a mask map can be generated and concatenated with the Mast3r features before passing into the another GS Head, similar to what GaussianTrim3R does after the Mask Generation Module. This ensures that pruning methods are applied before final Gaussian generation, making it a fairer baseline.
* The pruning method explored in this paper is limited to scene-level pruning. I suggest considering pixel-level pruning (Liu et al., EfficientGS: Streamlining Gaussian Splatting for Large-Scale High-Resolution Scene Representation, IEEE MM 2025), which ensures at least one Gaussian per ray and avoids holes.
* Ambiguous definition of “adaptive allocation” (line 455): where does the “adaptive” property come from? How is it related to disabling the training of the “Gaussian Head” in the “Without Gaussian Adaptation” ablation study?
* From an efficiency perspective: please report inference latency (fps or milliseconds). This would help readers better understand the pros and cons of zero-shot lightweight Gaussian generation.
* I found relatively fair baseline and recommend to compare with this: Fei et al., PixelGaussian: Generalizable 3D Gaussian Reconstruction from Arbitrary Views, arXiv

---

> ### Author Response · Authors · 2025-11-21
>
> We thank the reviewer for their constructive feedback
> # Q1:  Formatting and Presentation
>
> **Addressing textual corrections:**
>  Thank you for pointing out the typo and clarity issues. We have corrected the spelling oversight and revised the text.
>
> **Addressing figure clarity:**
>  We appreciate the detailed suggestions regarding Figure 4. We have updated the figure.
>
> # Q2: Performance at 0% Pruning
>
> Thank you for the suggestion. We have updated the manuscript accordingly. **Appendix Section E.1** now reports the full set of results at 0%, 5%, and 10% pruning for both ACID and RE10K, enabling a direct comparison between the unpruned backbone and the pruned variants. As shown in the updated tables (Table 14 and 15), existing pixel-aligned feed-forward methods (e.g., NoPoSplat) degrade sharply even under minor pruning. This confirms the well-known fragility of dense, pixel-aligned Gaussians to the removal of points, highlighting their lack of flexibility. In contrast, GaussianTrim3R maintains remarkably stable quality across 0–10% pruning, with a variation of less than **0.1 dB** in PSNR. Although our 0% performance is slightly lower than the dense backbone, it is expected because our GS Head is optimized for a range of pruning levels rather than only the dense prediction. The model is significantly more robust under any non-trivial pruning scenario. This stability is a direct consequence of our pruning-aware training and flexible representation, which redistributes representational capacity across the retained Gaussians rather than collapsing under sparsification. Our method is designed for scenarios where pruning is necessary to meet memory or rendering speed constraints. When no pruning is required, the base model can simply be used for full prediction. GaussianTrim3R enables continuous, real-time control over the quality efficiency tradeoff without per-scene optimization, particularly excelling in the moderate-to-severe pruning regime where practical gains are largest.
>
> ### ACID Dataset  0%, 5%, 10% Pruning Results
>
> | Method              | Pruning | PSNR   | LPIPS  | SSIM  |
> |---------------------|---------|--------|--------|-------|
> | NoPoSplat           | 0%      | 25.999 | 0.190  | 0.782 |
> | NoPoSplat (EAGLES)  | 5%      | 22.505 | 0.275  | 0.676 |
> | NoPoSplat (EAGLES)  | 10%     | 21.154 | 0.318  | 0.644 |
> | **Ours**            | 0%      | 22.843 | 0.272  | 0.662 |
> | **Ours**            | 5%      | 22.815 | 0.273  | 0.661 |
> | **Ours**            | 10%     | 22.787 | 0.274  | 0.659 |
>
>
> # Q3: Additional Baselines
>
> Our original baselines follow the standard post-hoc pruning paradigm adopted by current 3DGS pruning pipelines (EAGLES, LightGaussian, PUP-3DGS). These methods prune only after the full set of Gaussians has been predicted and are not structurally integrated into the feed-forward architecture. Creating a mask for pruning before Gaussian prediction, as suggested, requires retraining MASt3R with an auxiliary GS Head conditioned on a pruning mask, essentially reconstructing the core mechanism of GaussianTrim3R. Therefore, such a design does not constitute an independent baseline but rather a partial reimplementation of our method.
>
> We implemented the suggested “contextual-mask” variant (reported in **Appendix B** of the updated manuscript). This baseline predicts dense Gaussians, computes pruning masks via an existing pruning technique, and then uses these masks to supervise a second GS Head. As expected, this two-step variant performs better than the purely post-hoc baselines from the main paper because the contextual mask provides extra flexibility to the GS Head during attribute prediction.
>
> Despite this stronger baseline, GaussianTrim3R consistently outperforms all variants across all pruning strengths and on both ACID and RE10K (**Tables 9 and 10** in the appendix). The key reason is that our pruning strategy is texture-aware: it identifies structurally coherent regions for pruning and trains the GS Head to adaptively reallocate representational capacity to the retained Gaussians. In contrast, the contextual-mask baseline still requires generating all Gaussians upfront and then pruning them, making it a two-step pipeline rather than a pruning-aware feed-forward design introduced in our method. Thank you for this insightful suggestion. We have incorporated this additional baseline in the manuscript for comprehensive comparison.

---

> > ### Comment · Reviewer_D4Jq · 2025-11-21
> > **Thanks for the additional analysis.**
> >
> > Dear authors, I appreciate your sincere effort in clarifying my concerns and questions. I have the following comments and questions below.
> >
> > **Q1: Formatting and Presentation**
> > Thanks for addressing my suggestions. Figure 4 is now clearer to me. For Figure 6, could you also address my concern? I noticed that I referred to the wrong figure in my original comment: "In ~~Figure 4~~ Figure 6, I suggest representing the distributions using blue and yellow line curves ..." This is a very minor issue, but I suggest improving it to better convey the authors’ idea.
> >
> > **Q4: Comparison with EfficientGS**
> > The authors’ analysis of EfficientGS is mostly valid, except for one aspect. While LightGaussian evaluates importance over all images (global influence), EfficientGS does so within each pixel. They select the top-K Gaussians to be retained for each pixel, as represented in Eq. 8 of their paper: $weight^g_p \le sort({-weight^j_p,  g_j \in N_p})[K]$,
> >
> > where g and g_j are one of Gaussian group intersecting the camera ray of p, and p is the pixel index. By selecting the top-K Gaussians that are dominant for each pixel, they ensure minimal performance degradation from pruning. Their pixel-level selection is somewhat similar to GaussianTrim3R in that both methods address holes after pruning, which is why I suggested EfficientGS as another baseline.
> >
> > **Q6: Reporting Inference Latency**
> > In Table 12 of the revised paper, the authors provide a latency analysis:
> >
> > Table 12: Inference latency comparison across methods ...
> > | Method                          | Time (s) | PSNR (13K) |
> > |---------------------------------|----------|------------|
> > | Baselines (with finetuning)     | 6.58     | 9.626      |
> > | Baselines (without finetuning)  | 2.03     | 7.500      |
> > | Ours                            | 2.53     | 17.845     |
> >
> > When comparing *GaussianTrim3R* with *Baselines (without finetuning)*, *GaussianTrim3R* consumes 24.6% more time while achieving a substantial PSNR improvement of 10.345 dB, indicating an **efficient trade-off between latency and quality**. My follow-up question is: where does the additional 0.5 seconds of latency come from compared to the *Baselines*? I assume it might be due to the *Generation Module*, but I am not certain.

---

> ### Author Response · Authors · 2025-11-21
>
> ### ACID Dataset — Additional Baseline Comparison at 78K / 52K / 13K
>
> | Method        | PSNR (78K) | LPIPS (78K) | SSIM (78K) | PSNR (52K) | LPIPS (52K) | SSIM (52K) | PSNR (13K) | LPIPS (13K) | SSIM (13K) |
> |---------------|------------|--------------|-------------|------------|--------------|-------------|------------|--------------|-------------|
> | **EAGLES**        | 18.747     | 0.480        | 0.512       | 18.716     | 0.488        | 0.514       | 16.514     | 0.547        | 0.463       |
> | **LightGaussian** | 19.439     | 0.453        | 0.519       | 19.176     | 0.476        | 0.507       | 14.333     | 0.572        | 0.356       |
> | **PUP-3DGS**      | 17.912     | 0.555        | 0.503       | 17.193     | 0.578        | 0.492       | 13.177     | 0.662        | 0.384       |
> | **Ours**          | **22.549** | **0.299**    | **0.640**   | **22.310** | **0.310**    | **0.627**   | **18.476** | **0.379**    | **0.553**   |
>
>
> Q4: Comparison with EfficientGS: Streamlining Gaussian Splatting for Large-Scale High-Resolution Scene Representation, IEEE MM 2025
>
> **Clarifying the relationship between EfficientGS and our existing baselines:**
>  We appreciate the reviewer’s suggestion to compare with EfficientGS. EfficientGS belongs to the family of post-hoc, optimization-based pruning frameworks, similar in spirit to LightGaussian, which we have already included in our experimental comparison.
> EfficientGS operates by enforcing at least one Gaussian per pixel/ray through significance-based pruning. This strategy is conceptually aligned with LightGaussian, which also computes global significance scores (based on volume, opacity, and transmittance) and prunes Gaussians ranked as least important. Both methods follow the same philosophy:
>
> - Compute importance weights after 3D Gaussian optimization,
> - Prune Gaussians with low global influence,
> - Run additional refinement/finetuning iterations to adapt the remaining Gaussians.
>
> Our results already show that the LightGaussian based pruning technique degrade sharply when applied to feed-forward sparse-view settings for both with and without finetuning. Results for LightGaussian are included in Table 1 and Table 2 of the main paper and corresponding results in Figure 4 and Figure 5. Because EfficientGS uses the same post-optimization pruning mechanism (based on significance scores and per-ray coverage), its behavior under pruning would follow the same trend. Hence, the performance of EfficientGS will be similar to that of LightGaussian.
>
> # Q5: Clarifying “Adaptive Allocation”
>
> **Adaptive property**
> In GaussianTrim3R, adaptive allocation refers to the GS Head’s learned ability to modify the attributes of the remaining Gaussians after pruning, specifically their scale, opacity, and orientation, so that they can compensate for removed neighbors and preserve scene fidelity. During training, the GS Head observes masked-out regions produced by varying pruning ratios and learns to redistribute representational capacity accordingly. This enables the model to avoid holes and blobby artifacts even under aggressive pruning.
>
> **Relation to the “Without Gaussian Adaptation” ablation**
>  When we disable training of the GS Head (the Without Gaussian Adaptation ablation), the network is forced to use the frozen Gaussian attributes from the pretrained backbone. These Gaussians cannot adapt their scale or opacity to fill missing regions, which results in visible holes and degraded PSNR/SSIM. This effect is illustrated in Fig. 6 of the main paper and discussed in line 409. These results directly demonstrate that adaptive adjustment beyond simple masking is essential for maintaining reconstruction quality under pruning.
>
> # Q6: Reporting Inference Latency
>
> We have added detailed inference-time measurements in **Appendix C** of the revised manuscript (highlighted in blue for easier identification). The results include FPS values across a wide range of pruning strengths in the InstantSplat sparse-view setting.
>
> **Key observation:**
>  As pruning strength increases, the number of Gaussians to rasterize decreases, producing a consistent and monotonic improvement in FPS. This empirically validates that GaussianTrim3R enables practical, real-time control over rendering speed without requiring any per-scene optimization.
> ### FPS
> | Pruning Strength | 0%    | 40%   | 50%   | 60%   | 70%   | 80%   |
> |------------------|--------|--------|--------|--------|--------|--------|
> | **FPS**          | 191.4 | 226.7 | 236.3 | 255.0 | 281.4 | 301.3 |

---

> ### Author Response · Authors · 2025-11-21
>
> # Q7: Comparison with pixelGaussian: Generalizable 3D Gaussian Reconstruction from Arbitrary Views
>
> pixelGaussian requires **300K** iterations on **8× A6000 GPUs** for training, and **pretrained checkpoints are currently unavailable**, making direct evaluation infeasible. Furthermore, once pixelGaussian’s pretrained checkpoints are released, we will include a baseline using its feed-forward backbone for completeness.
>
> It is designed for generalizable 3D reconstruction from arbitrary numbers of input views, relying on a computationally heavy pipeline that densifies and iteratively refines Gaussians via the **Iterative Gaussian Refiner (IGR)** module. This refinement requires multiple deformable-attention iterations, making inference significantly more **expensive** and not suitable for **controllable pruning** or **inference time use**. In contrast, our work targets  controllable feed forward pruning via texture aware pruning rather than opacity thresholding without iterative refinement or per-scene optimization. We use a lightweight **DPT-based GS Head** to predict adapted Gaussians in a single forward pass, avoiding the computationally expensive multi-stage refinement used in PixelGaussian.
>
> Our manuscript provides an extensive evaluation (Table 1 and Table 2) across stronger and more relevant baselines that combine state-of-the-art feed-forward Gaussian predictors (PixelSplat, MVSplat, NoPoSplat) with state-of-the-art pruning methods (EAGLES, LightGaussian, PUP-3DGS). These constitute the correct baseline family for pruning-based efficiency methods. pixelGaussian belongs instead to a full-reconstruction paradigm and is not a controllable pruning or compression method.

---

> ### Author Response · Authors · 2025-11-23
>
> We thank the reviewer for their feedback and for the quick follow-up.
>
> # Q1: Formatting and Presentation:
>
> Thank you for the clarification. We have updated Figure 6 accordingly, replacing the overlapping filled regions with the suggested blue and yellow line-curve representations.
>
> # Q4: Comparison with EfficientGS
> We appreciate the suggestion. We have updated the main paper as follows:
> - Table 1 and Table 2 (main paper) now include EfficientGS as an additional pruning baseline.
> - Figure 4 has been updated to include EfficientGS results for both finetuned and non-finetuned settings on ACID and RE10K.
> - Figure 5 has been updated to include EfficientGS results.
> - Tables 16 and 17 in the Appendix now include EfficientGS as an additional pruning baseline.
>
> # Q6: Reporting Inference Latency
>
> The additional ~50 ms of latency compared to the baseline (without finetuning) comes primarily from the Mask Generation Module (specifically from SuperCluster creation).

---

> > ### Author Response · Authors · 2025-11-25
> > **Gentle Reminder**
> >
> > Dear Reviewer D4Jq
> >
> > With about a week remaining in the discussion period, we wanted to kindly invite you to take another look at our rebuttal and the updated results we have added based on your valuable feedback.
> >
> > We have addressed all of your comments in detail, updated several figures and tables, added additional baselines (including EfficientGS), reported 0% pruning results, and included full latency/FPS analyses. We would be very grateful if you could share any further thoughts, or let us know if there is anything else we can clarify.
> >
> > Thank you again for your time and for helping us improve the paper.
> >
> > Best regards,
> >
> > Authors

---

### Author Response · Authors · 2025-11-21
**Brief Summary of Paper Revisions**

We sincerely thank all reviewers for their thoughtful and constructive feedback. The revised manuscript incorporates the following updates to address the raised concerns:

1. Line 089 updated from “doesnt” to “doesn’t”.

2. **Figure 4** Now includes a clearer comparison using results from the same scene under both finetuned and non-finetuned settings. Also it includes an additional comparison of EfficientGS using results from the same scene under both finetuned and non-finetuned settings.

3. **Figure 5** Now includes additional comparison of EfficientGS for both ACID and RE10K dataset.

4. **Figure 6** is replaced with line plot instead of histogram for better understanding.

5. **Table 1 and Table 2** contains additional baseline (EfficientGS: Streamlining Gaussian Splatting for Large-Scale High-Resolution Scene Representation).

6. **Line 390-391** now mentions another pruning method (EfficientGS: Streamlining Gaussian Splatting for Large-Scale High-Resolution Scene Representation).

7. Ablation Studies in **Appendix Section A.4 (and Table 5)** now reports additional ablations on the RE10K dataset across multiple pruning strengths.

8. Additional Baselines in **Appendix Section B (Tables 9 and 10)** introduces new baselines where pruning masks are derived from existing pruning pipelines and integrated into the GS Head.

9. FPS and Inference Time Analysis in **Appendix Section C (Table 11 and Fig. 9)** presents FPS improvements across pruning strengths, and Table 12 reports latency measurements.

10. Extension to Multi-View Reconstruction using InstantSplat in **Appendix Section D (Table 13)** evaluates our pruning strategy within InstantSplat for 3-view, 4-view, and 10-view inputs.

11. 0%–10% Pruning Behavior in **Appendix Section E.1 (Table 15 and Fig. 10)** reports detailed results at 0%, 5%, and 10% pruning, highlighting our method’s stability compared to baselines.

12. Table 16 and Table 17 (in **Appendix**) contains additional results with another pruning method(EfficientGS: Streamlining Gaussian Splatting for Large-Scale High-Resolution Scene Representation)

All changes introduced in the appendix are highlighted in **blue text** for easy identification.

We hope the revised paper can effectively address reviewer's concerns. Please let us know if there are any further suggestions.

---

### Meta-Review · Area_Chair_NiZK · 2026-01-07

**Summary:**

The reviewers raised some concerns, including fairness in the baseline evaluation and comparisons (D4Jq, RPAH), ambiguities in the text (D4Jq), non-adaptiveness of k (qQcZ), more comparisons (KvJ4) and time complexity (RPAH), failure cases not sufficiently emphasized (RPAH) and missing evaluation on scenes with high-frequency texture (RPAH). The evaluation is marginally positive for three reviewers out of four.

**Reviewer Concerns:**

The authors tried to address the abovementioned point through a long rebuttal, providing more comparisons and clarifications to some of the raised points.

The major standing point lies in the fairness of the proposed comparisons. Indeed, as shown in the Q2 for D4Jq, the proposed method essentially seemingly obtains fewer Gaussians without allowing for good reallocation (there is a 3dB gap-in worsening- at 0% pruning), and the comparisons with the other methods could be misleading. It is agreed that the method targets explicitly fewer Gaussians, but this performance gap is a major negative downside not emphasized. This potentially raises other questions related to hyperparameters choice and impact (like the choice of k, ablated for the 78k and the 13k Gaussian cases in appendix A.1) realistically leads to different outcomes for 0% (131k, Table 14). Therefore, in such a sense, the ablation analysis feels incomplete, and the weakness raised by qQcZ still stands.

Besides, missing evaluation in the critical case with high-frequency texture, raised by RPAH, further highlights an aspect of the work is incomplete with.

After a thorough discussion with the SAC, it was agreed that the work, despite its merits, for the abovementioned reasons, is not ready to be accepted at ICLR.

**Reviewer Scores:**

Despite what indicated by the authors in the comments, it is felt that the evaluation could have stayed overall borderline. The merits of the paper are not under discussion, but the whole comparison setting and missing analyses naturally raising after rebuttal question the paper's readiness.

---

### Decision · Program_Chairs · 2026-01-26

Reject